# StructComp: Substituting propagation with Structural Compression in Training Graph Contrastive Learning

**Shengzhong Zhang**
Fudan University, Shanghai, China
`szzhang17@fudan.edu.cn`

**Wenjie Yang**
Fudan University, Shanghai, China
`yangwj22@m.fudan.edu.cn`

**Xinyuan Cao**
Georgia Institute of Technology, Midtown, USA
`xcao78@gatech.edu`

**Hongwei Zhang**
Fudan University, Shanghai, China
`hwzhang22@m.fudan.edu.cn`

**Zengfeng Huang**[*]
Fudan University, Shanghai, China
`huangzf@fudan.edu.cn`

## Abstract

Graph contrastive learning (GCL) has become a powerful tool for learning graph data, but its scalability remains a significant challenge. In this work, we propose a simple yet effective training framework called Structural Compression (Struct-Comp) to address this issue. Inspired by a sparse low-rank approximation on the diffusion matrix, StructComp trains the encoder with the compressed nodes. This allows the encoder not to perform any message passing during the training stage, and significantly reduces the number of sample pairs in the contrastive loss. We theoretically prove that the original GCL loss can be approximated with the contrastive loss computed by StructComp. Moreover, StructComp can be regarded as an additional regularization term for GCL models, resulting in a more robust encoder. Empirical studies on various datasets show that StructComp greatly reduces the time and memory consumption while improving model performance compared to the vanilla GCL models and scalable training methods.

## 1 Introduction

Graph neural networks (GNNs) (Kipf & Welling, 2017; Velickovic et al., 2018; Chen et al., 2020b; Liu et al., 2020) provide powerful tools for analyzing complex graph datasets and are widely applied in various fields such as recommendation system (Cai et al., 2023), social network analysis (Zhang et al., 2021a), traffic flow prediction (Wang et al., 2020), and molecular property prediction (Alex Fout & Ben-Hur, 2019). However, the scalability limitation of GNNs hampers their extensive adoption in both industrial and academic domains. This challenge is particularly pronounced within the realm of unsupervised graph contrastive learning (GCL). Compared to the research on the scalability of supervised GNNs (Hamilton et al., 2017; Chen et al., 2018c;b; Zou et al., 2019; Cong et al., 2020; Ramezani et al., 2020; Markowitz et al., 2021; Zeng et al., 2020; Chiang et al., 2019; Zeng et al., 2021), there is little attention paid to the scalability of GCL (Wang et al., 2022; Zheng et al., 2022), and there is no universal framework for training various GCL models.

The scalability issue of GCL mainly has two aspects: Firstly, the number of nodes that need to be computed in message passing grows exponentially. Secondly, GCL usually requires computation of a large number of sample pairs, which may require computation and memory quadratic in the number of nodes. At the same time, the graph sampling (Zeng et al., 2020; Chiang et al., 2019; Zeng et al., 2021) and decoupling technology (Wu et al., 2019; Zhu & Koniusz, 2021) used for

---

[*]Corresponding author

supervised GNN training are not applicable to GCL. Graph sampling might affect the quality of positive and negative samples, thereby reducing the performance of the model. As GCL usually involves data augmentation, the decoupling method that precomputes the diffusion matrix is not feasible.

In order to solve the two aforementioned problems simultaneously, we use a sparse assignment matrix to replace message passing, which is a low-rank approximation of the diffusion matrix. Utilizing the assignment matrix, we can compute the mixed node features, which contain all the local node information and can be regarded as the community center feature. By controlling the similarity of the community center embeddings, we can make node embeddings in similar communities close to each other, and node embeddings in dissimilar communities distant from each other. Since the number of sample pairs needed for computation by mixed nodes is significantly less than that for full nodes, and the encoder no longer computes message passing, the computational resources required for training are greatly saved.

Specifically, we propose an extremely simple yet effective GCL training framework, called Structural Compression (StructComp). This framework is applicable to various single-view GCL models and multi-view GCL models. During the training process, the GNN encoder does not need to perform message passing, at which point the encoder can be regarded as a MLP. Our model takes the compressed features as input and trains in the same way as the corresponding GCL model (i.e., the same loss function, optimization algorithm). In the inference process, we take the complete graph structure information and node features as input, and use the GNN encoder to obtain the embedding representations of all nodes.

Our contributions are summarized as follows:

1. We propose a novel GCL training framework, StructComp. Motivated by a low-rank approximation of the adjacency matrix, StructComp significantly improves the scalability of GCLs by substituting message-passing with node compression. StructComp trains MLP encoder on these mixed nodes and later transfers parameters to GNN encoder for inference.

2. We customize a data augmentation method specifically for StructComp, making StructComp adaptive to both single-view and multi-view GCL models. To the best of our knowledge, StructComp is the first unified framework designed specifically for GCL training.

3. We theoretically guarantee that the compressed contrastive loss can be used to approximate the original graph contrastive loss. And we prove that our method introducing an extra regularization term into the scalable training, which makes the model more robust.

4. We empirically compare StructComp with full graph training and other scalable training methods under four GCL models. Experimental results on seven datasets demonstrate that StructComp improves the GCL model's performance, and significantly reduces memory consumption and training time.

## 2 PRELIMINARIES

**Notation.** Consider an undirected graph $G = (A, X)$, where $A \in \{0, 1\}^{n \times n}$ represents the adjacency matrix of $G$, and $X \in \mathbb{R}^{n \times d}$ is the feature matrix. The set of vertices and edges is represented as $V$ and $E$, with the number of vertices and edges given by $n = |V|$ and $m = |E|$, respectively. The degree of node $v_i$ denoted as $d_i$. The degree matrix $D$ is a diagonal matrix and its $i$-th diagonal entry is $d_i$.

**Graph neural network encoders**. The GNN encoders compute node representations by aggregating and transforming information from neighboring nodes. One of the most common encoders is the Graph Convolutional Network (GCN) (Kipf & Welling, 2017), and its propagation rule is defined as follows:

$$H^{(l+1)} = \sigma \left( \widetilde{D}^{-\frac{1}{2}} \widetilde{A} \widetilde{D}^{-\frac{1}{2}} H^{(l)} W^{(l)} \right),$$

(1)

where $\widetilde{A} = A + I$, $\widetilde{D} = D + I$ and $W^{(l)}$ is a learnable parameter matrix. GCNs consist of multiple convolution layers of the above form, with each layer followed by an activation $\sigma$ such as ReLU.

**Graph contrastive learning**. Graph contrastive learning is an unsupervised graph representation learning method. Its objective is to learn the embeddings of the graph by distinguishing between

similar and dissimilar nodes. Common methods of graph contrastive learning can be divided into two types: single-view graph contrastive learning (Zhang et al., 2020; Zhu et al., 2021a) and multi-view graph contrastive learning (Zhu et al., 2020; 2021b; Zhang et al., 2021b; Zheng et al., 2022).

In single-view graph contrastive learning, positive and negative sample pairs are generated under the same view of a graph. In this case, the positive samples are typically pairs of adjacent or connected nodes, while the negative samples are randomly selected pairs of non-adjacent nodes. Then, through a GNN encoder and a contrastive loss function, the algorithm learns to bring the embedding vectors of positive sample pairs closer and push the embedding of negative sample pairs further apart. The common single-view contrastive loss function (Hamilton et al., 2017) of node $u$ is as follows:

$$\mathcal{L}(u) = -\log(\sigma(z_u^T z_v)) - \sum_{k=1}^{K} \log(\sigma(-z_u^T z_k)).$$ (2)

Here, node $v$ is the positive sample of node $u$, node $k$ is the negative sample of node $u$, and $K$ represents the number of negative samples.

Multi-view graph contrastive learning uses different views of the graph to generate sample pairs. These views can be contracted by different transformations of the graph, such as DropEdge (Rong et al., 2020) and feature masking (Zhu et al., 2020). We generate the embeddings of the nodes, aiming to bring the embedding vectors of the same node but from different views closer, while pushing the embedding vectors from different nodes further apart. The common multi-view contrastive loss function of each positive pair $(u, v)$ is as follows:

$$\mathcal{L}(u,v) = \log \frac{e^{\phi(z_u, z_v)/\tau}}{e^{\phi(z_u, z_v)/\tau} + \sum_{k \neq u, k \in G_1} e^{\phi(z_u, z_k)/\tau} + \sum_{k \neq u, k \in G_2} e^{\phi(z_u, z_k)/\tau}}.$$ (3)

Here $u$ and $v$ represent the same node from different views, $\phi$ is a function that computes the similarity between two embedding vectors. $G_1$ and $G_2$ are two different views of the same graph. $\tau$ is temperature parameter.

## 3 STRUCTURAL COMPRESSION

### 3.1 MOTIVATION

To reduce the training complexity of GCLs, we start with a low-rank approximation $C$ of the adjacency matrix $\hat{A}^k$, such that $\hat{A}^k = CC^T$. Although the complexity of matrix multiplication is significantly reduced by the approximation, the actual training time will not decrease due to the dense nature of $C$. Moreover, the amount of negative pairs needed for the contrastive learning remains $O(n^2)$. To address the above issues simultaneously, we introduce a sparse constraint for the low-rank approximation and force $C$ to be a graph partition matrix $P' \in \mathbb{R}^{n \times n'}$ ($P'_{ij} = 1$ if and only if the node $i$ belongs to cluster $j$), where $n'$ is the number of clusters in the partition. Using $P'P^T X$ to approximate $\hat{A}^k X$ (where $P$ is the row-normalized version of $P'$), a key advantage is that nodes in the same cluster share the same embedding, and $P^T X$ contains all the information needed to compute the loss function. Therefore, we only compute $P^T X \in \mathbb{R}^{n' \times d}$: a "node compression" operation, where nodes in the same cluster are merged together. Since nodes in the same cluster share their embeddings, performing contrastive learning on these compressed nodes is equivalent to that on the nodes after the low-rank propagation (i.e., $P'P^T X$). Thus, the number of negative pairs reduced to $n'^2$. Moreover, the complexity of matrix multiplication is now down to $O(n)$ while persevering the sparsity. In a nutshell, two major challenges for scalable GCL in Section 1 can be solved simultaneously by our method.

To generate the graph partition matrix $P$, we need to solve the following optimization problem:

$$\begin{aligned} \text{minimize} \quad & \|P'P^T - \hat{A}^k\|, \\ \text{subject to} \quad & P' \in \{0, 1\}^{n \times n'}, P'1_{n'} = 1_n. \end{aligned}$$ (4)

Intuitively, $P'P^T$ is a normalized graph that connects every pair of nodes within a community while discarding all inter-community edges. Thus, $\|P'P^T - \hat{A}\|$ equals the number of inter-community

edges plus the number of disconnected node pairs within communities. Minimizing the former is the classic minimum cut problem, and minimizing the latter matches well with balanced separation, given the number of nodes pair grows quadratically. These objective aligns well with off-the-shelf graph partition methods, so we directly utilize METIS to produce $P$.

From the perspective of spatial domain, we always hope to obtain an embedding $f(\hat{A}^k XW)$ of the following form: embeddings of nodes of the same class are close enough, while embeddings of nodes of different classes are far apart. Intuitively, this goal can be simplified to the class centers of the various class node embeddings being far apart, and the similarity of nodes within the same class being as high as possible. In other words, we can use the community center $f(P^T XW)$ as the embedding that needs to be computed in the loss function, instead of using all nodes in the community for computation. On the other hand, if the embeddings of nodes within the community are identical, we no longer need to compute the similarity of nodes within the community. Considering these, it is natural to use $P^T X$ in place of $\hat{A}^k X$ to compute the loss function. Moreover, since $P$ is solved based on graph partitioning, the result of the graph partitioning can facilitate the construction of positive and negative samples. The idea of using structural compression as a substitute of message-passing can be extended to a multi-layer and non-linear GNN, which is shown in Appendix A.1.

### 3.2 FRAMEWORK OF STRUCTCOMP

**Preprocessing**. We carry out an operation termed "node compression". Based on the above analysis, we use the METIS algorithm to obtain the graph partition matrix, and then take the mean value of the features of the nodes in each cluster as the compressed feature, i.e., $X_c = P^T X$. After computing the compressed features, we also construct a compressed graph, i.e., $A_c = P^T AP$. Each node in $A_c$ represents a cluster in the original graph, and the edges represent the relationships between these clusters. $A_c$ is only used when constructing the contrastive loss and is **not** involved in the computations related to the encoder. To make our StructComp adaptive to different types of GCL models, we carefully design some specific modules for single-view and multi-view GCLs, respectively.

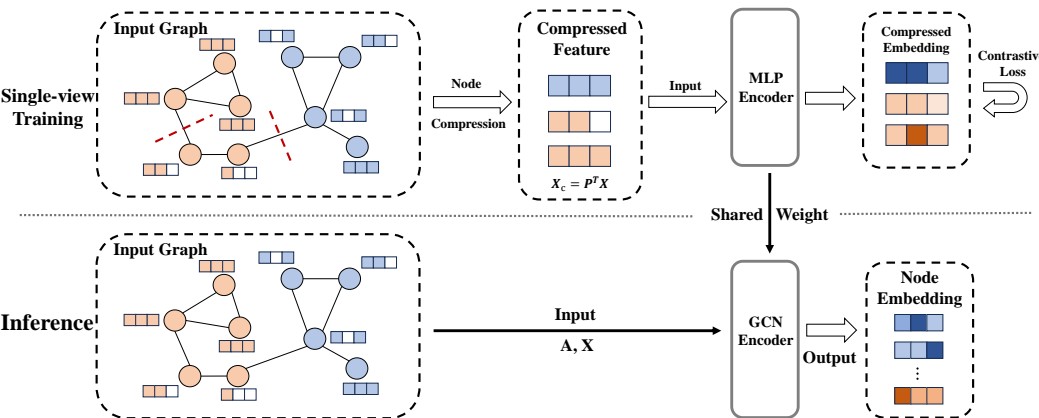

Figure 1: The overall framework of single-view StructComp.

**Single-view StructComp.** In single-view graph contrastive learning, we use the preprocessed compression features $X_c$ as input, and replace the GNN encoders with MLP encoders. For instance, a two-layer neural network and embedding can be represented as follows:

$$Z_c = \sigma(\sigma(X_c W_1)W_2) \tag{5}$$

We proposed to sample positive and negative pairs based on the compressed graph $A_c$ instead of $A$. One additional advantages of using the compressed graph over the original graph is that it significantly improves the accuracy of negative pairs sampling. For instance, since highly connected nodes are compressed together, they are not able to be selected as negative pairs. Then we use the same loss function and optimization algorithm as original GCL models to optimize the single-view contrastive learning loss $L(Z_c)$. Figure 1 shows the flow chart of single-view StructComp.

Once the model is adequately trained, we transition to the inference phase. We revert the changes made during the training phase by replacing the MLP encoder back to the GNN encoder. Then, we input the complete graph structure information and node features to generate the embeddings for all nodes, as detailed below:

$$Z = \sigma(\hat{A}\sigma(\hat{A}XW_1)W_2). \tag{6}$$

**Multi-view StructComp.** In multi-view contrastive learning, we need to compare two perturbed views of the graph. This requires us not only to compress the node features but also to apply data augmentation to these compressed features. However, traditional data augmentation methods such as DropEdge are not applicable in StructComp, as there is no longer an $A$ available for them to perturb during training. To fill this gap, we introduce a new data augmentation method called 'DropMember'. This technique offers a novel way to generate different representations of compressed nodes, which are essential for multi-view contrastive learning under StructComp.

The DropMember method is implemented based on a predetermined assignment matrix P. For each node in the compressed graph, which represents a community, we randomly drop a portion of the nodes within the community and recalculate the community features. Formally, for each cluster $j$ in the augmented $X'_c$, we have:

$$x'_j = \frac{1}{s'} \sum_{i=1}^{s} m_i x_i. \tag{7}$$

Here, $s$ represents the number of nodes contained in cluster $j$, $m_i$ is independently drawn from a Bernoulli distribution and $s' = \sum_{i=1}^{s} m_i$. By performing contrastive learning on the compressed features obtained after DropMember and the complete compressed features, we can train a robust encoder. The loss of some node information within the community does not affect the embedding quality of the community.

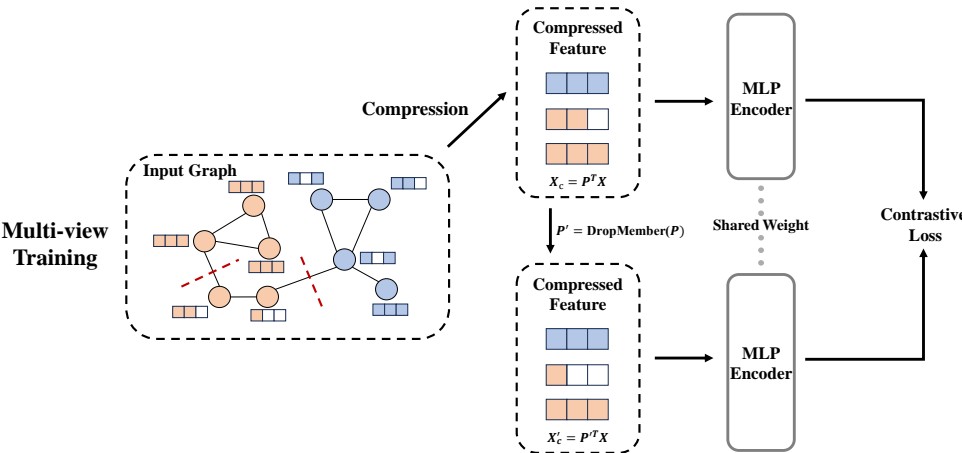

Figure 2: The training process of multi-view StructComp.

For the multi-view graph contrastive learning model, we need to compute the representations of two different views. Figure 2 shows the training process of multi-view StructComp. In our implementation, we use the complete $X_c$ and the perturbed $X'_c$ after DropMember as two different views. The embeddings of the two views are as follows:

$$Z_c = \sigma(\sigma(X_cW_1)W_2), \quad Z'_c = \sigma(\sigma(X'_cW_1)W_2). \tag{8}$$

We use the same loss function and optimization algorithm as original multi-view GCL models to optimize the contrastive loss $\mathcal{L}(Z_c, Z'_c)$. Once the model is trained, the inference process of multi-view StructComp is the same as that of single-view StructComp.

# 4 THEORY ANALYSIS OF STRUCTCOMP

## 4.1 THE EQUIVALENCE OF THE COMPRESSED LOSS AND THE ORIGINAL LOSS

In this section, we demonstrate that the contrastive loss on the original graph is close to the sum of the compressed contrastive loss and the low-rank approximation gap. In other words, if the low-rank approximation in section 3.1 is properly satisfied, we can estimate the original GCL loss using the compressed contrastive loss.

Here we consider the Erdős-Rényi model, denoted as $G(n, p)$, where edges between $n$ vertices are included independently with probability $p$. We use $\| \cdot \|_2$ to represent $l_2$ norm and $\| \cdot \|_F$ to represent Frobenius norm. Additionally, we denote the feature vector of each node as $X_i \in \mathbb{R}^d$. Then we can prove the following theorem, for simplicity, we only consider a one-layer message-passing and an unweighted node compression. We leave the proof and details of notations in Appendix A.2.

**Theorem 4.1.** *For the random graph $G(n, p)$ from Erdős-Rényi model, we construct an even partition $\mathcal{P} = \{S_1, \cdots, S_{n'}\}$. Let $f_G(X) = AXW$ be a feature mapping in the original graph and $f_\mathcal{P}(X) = P'^T XW$ as a linear mapping for the mixed nodes, where $W \in \mathbb{R}^{d \times d'}$. Then by conducting single-view contrastive learning, the contrastive loss for the original graph, denoted as $\mathcal{L}_G(W)$, can be approximated as the sum of the compressed contrastive loss, $\mathcal{L}_\mathcal{P}(W)$, and a term related to the low-rank approximation. Assume the features are bounded by $S_X := \max_i \|X_i\|_2$, we have*

$$|\mathcal{L}_G(W) - \mathcal{L}_\mathcal{P}(W)| \leq \|A - P' P'^T\|_F S_X \|W\|_2.$$

For a similar upper bound without the Erdős-Rényi graph assumption, please refer to Appendix A.3.

## 4.2 THE REGULARIZATION INTRODUCED BY STRUCTCOMP

Following Fang et al. (2023), we show that multi-view StructComp is equivalent to random masking on the message matrices $M$, where $M_{i,j} = \psi(h_i, h_j, e_{i,j})$ and $\psi$ is a function that takes the edges $e_{i,j}$ and the attached node representations $h_i, h_j$. First, the low rank approximation $||P'P - \hat{A}^k||$ is dropping the inter-cluster edges $E_{\text{drop}} = \{E_{i,j} | A_{i,j}^k = 1 \text{ and } S(i) \neq S(j)\}$, where $S(i)$ denote the cluster that node $i$ belongs to. And the latter is then equivalent to DropMessage $M_{\text{drop}} = \{M_i | \text{edge}(M_i) \in E_{\text{drop}}\}$, where $\text{edge}(M_i) \in E_{\text{drop}}$ indicates which edge that $M_i$ corresponds to. Our DropMember for the cluster $c$ is dropping $V_{\text{drop}}^c = \{X_i | \epsilon_i = 0 \text{ and } S(i) = c\}$. This is equivalent to $M_{\text{drop}} = \{M_i | \text{node}(M_i) \in \bigcup_c V_{\text{drop}}^c\}$. Then we have the following theorem:

**Theorem 4.2.** *Consider a no-augmentation InfoNCE loss,*

$$\mathcal{L}_{\text{InfoNCE}} = \sum_i \sum_{j \in \text{pos}(i)} [h_i^T h_j] + \sum_i \sum_{j \in \text{neg}(i)} [\log(e^{h_i^T h_i} + e^{h_i^T h_j})]. \tag{9}$$

*Optimizing the expectation of this with augmentation $\mathbb{E}[\tilde{\mathcal{L}}_{\text{InfoNCE}}]$ introduce an additional regularization term, i.e.,*

$$\mathbb{E}[\tilde{\mathcal{L}}_{\text{InfoNCE}}] = \mathcal{L}_{\text{InfoNCE}} + \frac{1}{2} \sum_i \sum_{j \in \text{neg}(i)} \phi(h_i, h_j) \text{Var}(\tilde{h}_i), \tag{10}$$

*where $\phi(h_i, h_j) = \frac{(e^{h_i^2} h_i^2 + e^{h_i h_j} h_j^2)(e^{h_i^2} + e^{h_i h_j}) - (e^{h_i^2} h_i + e^{h_i h_j} h_j)^2}{2(e^{h_i^2} + e^{h_i h_j})^2}$.*

Theorem 4.2 shows that, multi-view StructComp not only improves the scalability of GCLs training, but also introduces an additional regularization term into the InfoNCE loss. By optimizing the variance of the augmented representations, encoders trained with StructComp are more robust to minor perturbation. Please refer to the Appendix A.4 for more details.

# 5 RELATED WORK

**Scalable training on graph.** To overcome the scalability issue of training GNNs, most of the previous scalable GNN training methods use sampling techniques (Hamilton et al., 2017; Chen et al.,

2018c;b; Zou et al., 2019; Cong et al., 2020; Ramezani et al., 2020; Markowitz et al., 2021; Zeng et al., 2020; Chiang et al., 2019; Zeng et al., 2021), including node-wise sampling, layer-wise sampling, and graph sampling. The key idea is to train GNNs with small subgraphs instead of the whole graph at each epoch. However, graph sampling techniques are mainly used for training supervised GNNs and are not applicable to unsupervised GNNs, as it is difficult to guarantee the provision of high-quality positive and negative samples after sampling. Another direction for scalable GNNs is to simplify models by decoupling the graph diffusion process from the feature transformation. The diffusion matrix is precomputed, and then a standard mini-batch training can be applied (Bojchevski et al., 2020; Chen et al., 2020a; Wu et al., 2019). This preprocessing method is also not applicable to graph contrastive learning, as the adjacency matrix and feature matrix are perturbed in contrastive learning, which necessitates the repeated computation of the diffusion matrix, rather than only in preprocessing. Besides, methods represented by GraphZoom (Chen et al., 2018a; Liang et al., 2018; Deng et al., 2019) learn node embeddings on the coarsened graph, and then refine the learned coarsed embeddings to full node embeddings. These methods mainly consider graph structural information, only applicable to handling traditional graph embedding(Perozzi et al., 2014; Grover & Leskovec, 2016), but are not suitable for GCL models. Most importantly, these methods require a lot of time to construct the coarsened graph, and the coarse-to-refine framework inevitably leads to information loss.

See the Appendix B for a discussion of the related work on graph contrastive learning.

# 6 EXPERIMENT

## 6.1 EXPERIMENTAL SETUP

The results are evaluated on night real-world datasets (Kipf & Welling, 2017; Veličković et al., 2018; Zhu et al., 2021b; Hu et al., 2020), Cora, Citeseer, Pubmed, Amazon Computers, Amazon Photo, Ogbn-Arixv, Ogbn-Products and Ogbn-Papers100M. On small-scale datasets, including Cora, Citeseer, Pubmed, Amazon Photo and Computers, performance is evaluated on random splits. We randomly select 20 labeled nodes per class for training, while the remaining nodes are used for testing. All results on small-scale datasets are averaged over 50 runs, and standard deviations are reported. For Ogbn-Arixv, Ogbn-Products and Ogbn-Papers100M, we use fixed data splits as in previous studies Hu et al. (2020). More detailed statistics of the night datasets are summarized in the Appendix C.

We use StructComp to train two representative single-view GCL models, SCE (Zhang et al., 2020) and COLES (Zhu et al., 2021a), and two representative multi-view GCL models, GRACE (Zhu et al., 2020) and CCA-SSG (Zhang et al., 2021b). To demonstrate the effectiveness of StructComp, we compare the classification performance of the original models and StructComp trained models on small-scale datasets. For scalability on large graphs, we compare StructComp with three scalable training methods (i.e., Cluster-GCN (Chiang et al., 2019), Graphsaint (Zeng et al., 2020) and Graphzoom (Deng et al., 2019)). For all the models, the learned representations are evaluated by classifiers under the same settings.

The key hyperparameter of our framework is the number of clusters, which is set to [300, 300, 2000, 1300, 700, 20000, 25000, 5000] on night datasets, respectively. All algorithms and models are implemented using Python and PyTorch Geometric. More implementation details can be found in Appendix C. Additional discussions and experimental results are included in Appendix D.

## 6.2 EXPERIMENTAL RESULTS

**Performance on small-scale datasets.** Table 1 shows the model performance on small datasets using the full graph training and StructComp training. The results show that StructComp improves the performance of the model in the vast majority of cases, especially the multi-view GCL models. In the single-view GCL models, StructComp improves the average accuracy of SCE and COLES by 0.4% and 0.2%, respectively. In the multi-view GCL models, StructComp improves the average accuracy of GRACE and CCA-SSG by 2.6% and 1.6%, respectively. The observed performance improvement can be attributed to two main factors. First, StructComp constructs high-quality positive and negative pairs, e.g., it ensures that highly-connected nodes are not erroneously selected as

negative pairs in multi-view GCLs. Second, as mentioned in Section 4.2, StructComp implicitly introduces regularization into the contrastive learning process, resulting in a more robust encoder.

Table 1: Comparison between StructComp and full graph training across four GCL models on small datasets. The performance is measured by classification accuracy. "Ave $\Delta$" is the average improvement achieved by StructComp.

| Method | Cora | Citeseer | Pubmed | Computers | Photo | Ave $\Delta$ |
|---|---|---|---|---|---|---|
| SCE | 81.0±1.3 | 71.7±1.1 | 76.5±2.8 | 79.2 ±1.7 | 87.8 ±1.4 | |
| SCE$_{StructComp}$ | 81.6±0.9 | 71.5±1.0 | 77.2±2.9 | 79.7 ±1.7 | 88.2 ±1.4 | +0.4 |
| COLES | 81.7±0.9 | 71.2±1.2 | 74.6±3.4 | 79.5±1.6 | 88.5±1.4 | |
| COLES$_{StructComp}$ | 81.8±0.8 | 71.6±0.9 | 75.3±3.1 | 79.4±1.6 | 88.5±1.4 | +0.2 |
| GRACE | 78.5±0.9 | 68.9±1.0 | 76.1±2.8 | 76.2±1.9 | 85.1±1.6 | |
| GRACE$_{StructComp}$ | 79.7±0.9 | 70.5±1.0 | 77.2±1.4 | 80.6±1.5 | 90.0±1.1 | +2.6 |
| CCA-SSG | 79.2±1.4 | 71.8±1.0 | 76.0±2.0 | 82.7±1.0 | 88.7±1.1 | |
| CCA-SSG$_{StructComp}$ | 82.3±0.8 | 71.6±0.9 | 78.3±2.5 | 83.1±1.4 | 90.8±1.0 | +1.6 |

**Time and memory usage for small-scale datasets.** Table 2 shows the improvements in runtime and memory usage of each GCL model. StructComp saves the memory usage of the GCL models on the Cora, Citeseer, Pubmed, Computers, and Photo datasets by 5.9 times, 4.8 times, 33.5 times, 22.2 times and 57.1 times, respectively. At the same time, the training is speeded up by 1.8 times, 2.1 times, 17.1 times, 10.9 times and 21.1 times, respectively. This improvement is particularly evident when the dataset is large. The memory consumption for GRACE on the Computers dataset is even reduced by two orders of magnitude. These results strongly suggest that our method significantly reduces time consumption and memory usage while enhancing model performance.

Table 2: Time (s/epoch) and memory usage (MB) for GCL training on small-scale datasets. "Ave improvement" is the proportion of training resources used by StructComp to the resources used by full graph training.

| Method | Cora | | Citeseer | | Pubmed | | Photo | | Computers | |
|---|---|---|---|---|---|---|---|---|---|---|
| | Mem | Time | Mem | Time | Mem | Time | Mem | Time | Mem | Time |
| SCE | 82 | 0.003 | 159 | 0.004 | 1831 | 0.027 | 329 | 0.006 | 920 | 0.015 |
| SCE$_{StructComp}$ | 23 | 0.002 | 59 | 0.002 | 54 | 0.003 | 16 | 0.002 | 29 | 0.002 |
| COLES | 115 | 0.004 | 204 | 0.004 | 1851 | 0.033 | 378 | 0.015 | 1018 | 0.031 |
| COLES$_{StructComp}$ | 24 | 0.002 | 60 | 0.003 | 61 | 0.003 | 21 | 0.003 | 39 | 0.003 |
| GRACE | 441 | 0.017 | 714 | 0.025 | 11677 | 0.252 | 1996 | 0.106 | 5943 | 0.197 |
| GRACE$_{StructComp}$ | 37 | 0.009 | 72 | 0.009 | 194 | 0.009 | 59 | 0.008 | 54 | 0.008 |
| CCA-SSG | 132 | 0.010 | 225 | 0.011 | 825 | 0.123 | 1197 | 0.112 | 2418 | 0.210 |
| CCA-SSG$_{StructComp}$ | 38 | 0.006 | 71 | 0.005 | 85 | 0.006 | 41 | 0.005 | 40 | 0.005 |
| Ave improvement | 5.9× | 1.8× | 4.8× | 2.1× | 33.5× | 17.1× | 22.2× | 10.9× | 57.1× | 21.1× |

**Scalability on large graphs.** Table 3 and Table 4 show the results of StructComp on two large datasets Arxiv and Products, and compare them with three scalable training methods: Cluster-GCN, Graphsaint, and Graphzoom. We tested these training methods for four GCL models. Our method achieves the best performance on all GCL models on both datasets. The experimental results for Papers100M are listed in Appendix D.1. The models trained by our method achieve the highest accuracy, while also require significantly lower memory and time consumption compared to other models. These results highlight the effectiveness of our method in handling large-scale graph data.

Table 3: Performance and training consumption (time in s/epoch and memory usage in GB) on the Ogbn-Arxiv dataset. Each model is trained by StructComp and three scalable training frameworks.

| Method | SCE | | | COLES | | | GRACE | | | CCA-SSG | | |
|---|---|---|---|---|---|---|---|---|---|---|---|---|
| | Acc | Time | Mem | Acc | Time | Mem | Acc | Time | Mem | Acc | Time | Mem |
| Cluser-GCN | 70.4±0.2 | 10.8 | 4.2 | 71.4±0.2 | 13.0 | 4.2 | 70.1±0.1 | 3.5 | 17.3 | 72.2±0.1 | 2.2 | 1.3 |
| Graphsaint | 70.4±0.2 | 7.6 | 9.0 | 71.3±0.1 | 26.9 | 22.5 | 70.0±0.2 | 3.6 | 14.1 | 72.1±0.1 | 4.4 | 1.8 |
| Graphzoom | 70.6±0.2 | 0.04 | 9.8 | 70.0±0.3 | 0.08 | 14.3 | 68.2±0.3 | 3.9 | 13.7 | 71.3±0.2 | 1.0 | 3.4 |
| StructComp | **71.6**±0.2 | **0.03** | **1.8** | **71.8**±0.2 | **0.05** | **3.4** | **71.7**±0.2 | **1.2** | **11.7** | **72.2**±0.1 | **0.9** | **0.5** |

Table 4: Performance and training consumption (time in s/epoch and memory usage in GB) on the Ogbn-Products dataset. Each model is trained by StructComp and three scalable training frameworks.

| Method | SCE | | | COLES | | | GRACE | | | CCA-SSG | | |
|---|---|---|---|---|---|---|---|---|---|---|---|---|
| | Acc | Time | Mem | Acc | Time | Mem | Acc | Time | Mem | Acc | Time | Mem |
| Cluser-GCN | 74.6±0.2 | 196.8 | 8.0 | 75.2±0.2 | 239.1 | 8.0 | 75.2±0.1 | 35.1 | 16.2 | 74.1±0.3 | 37.1 | 5.7 |
| Graphsaint | 74.5±0.3 | 18.5 | 9.5 | 75.3±0.1 | 20.6 | 9.9 | 75.4±0.2 | 36.8 | 14.3 | 74.8±0.4 | 35.4 | 3.3 |
| Graphzoom | 60.6±0.5 | 0.06 | 8.8 | 68.1±0.4 | 0.1 | 13.2 | 61.0±0.4 | 11.1 | 13.1 | 68.6±0.3 | 4.5 | 10.0 |
| StructComp | **75.2**±0.1 | **0.05** | **2.7** | **75.5**±0.1 | **0.08** | **5.3** | **75.7**±0.1 | **4.0** | **12.0** | **75.8**±0.2 | **3.7** | **0.6** |

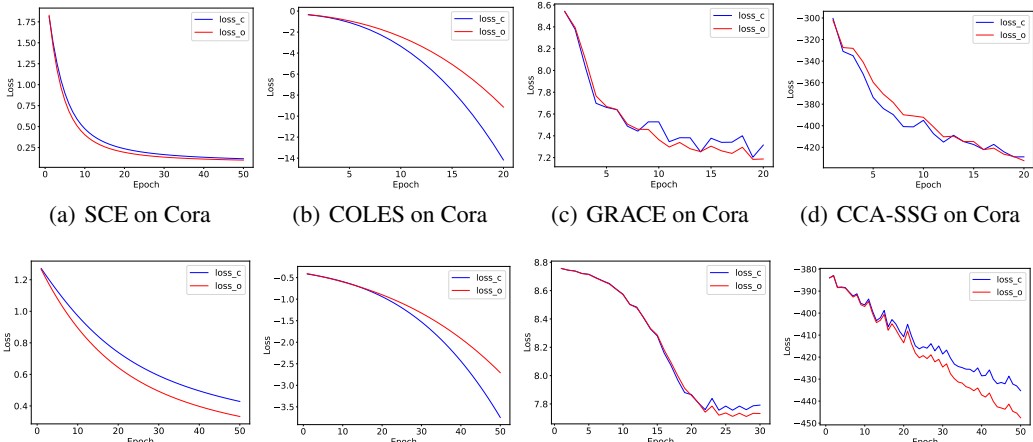

(a) SCE on Cora    (b) COLES on Cora    (c) GRACE on Cora    (d) CCA-SSG on Cora

(e) SCE on CiteSeer    (f) COLES on CiteSeer    (g) GRACE on CiteSeer    (h) CCA-SSG on CiteSeer

Figure 3: The trends of the original GCL loss and the loss that computed by StructComp-trained parameters. "loss_o" is $\mathcal{L}(A, X; W)$ and "loss_c" is $\mathcal{L}(A, X; U)$ where $U$ is trained with $\mathcal{L}(X_c; U)$.

**Comparison of loss trends.** To examine the equivalence of StructComp training and traditional GCL training, we plug the parameters $U$ trained with the compressed loss $\mathcal{L}(X_c; U)$ back into the GNN encoder and compute the complete loss function $\mathcal{L}(A, X; U)$. Figure 3 shows the trends of $\mathcal{L}(A, X; U)$ and the original loss $\mathcal{L}(A, X; W)$ trained with full graph. The behavior of the two losses matches astonishingly. This observation implies that, StructComp produces similar parameters to the traditional full graph GCL training.

**Effect of compression rate.** We study the influence of the compression rate on the performance of StructComp. We use StructComp to train four GCL models under different compression rates on Cora, Citeseer, and Pubmed. Figure 4 shows the performance change with respect to compression rates. We observed that when the compression rate is around 10%, the performance of the models is optimal. If the compression rate is too low, it may leads to less pronounced coarsened features, thereby reducing the effectiveness of the training.

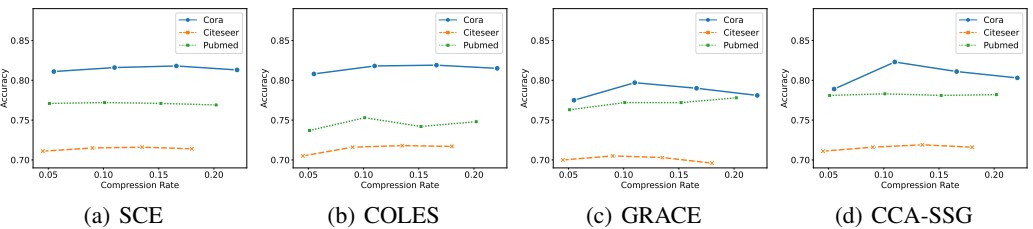

(a) SCE    (b) COLES    (c) GRACE    (d) CCA-SSG

Figure 4: The influence of the compression rate on the performance of StructComp.

## 7 CONCLUSION

In this paper, we introduce StructComp, a scalable training framework for GCL. StructComp is driven by a sparse low-rank approximation of the diffusion matrix. In StructComp, the message-passing operation is substituted with node compression, leading to substantial reductions in time and memory consumption. Theoretical analysis indicates that StructComp implicitly optimizes the original contrastive loss with fewer resources and is likely to produced a more robust encoder.

ACKNOWLEDGMENTS

This work is supported by National Natural Science Foundation of China No. U2241212, No. 62276066.

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

# A PROOF DETAILS

## A.1 THE NON-LINEAR EXTENSION FOR EQUATION 4

We explain how to extend the results to non-linear deep models below. Equation 4 provides the motivation of structural compression on a linear GNN (which can also be considered as an approximation to one layer in a multi-layer non-linear GNN). The analysis can be extended to non-linear deep GNNs. For instance, given a two-layer non-linear GCN $\sigma(\hat{A}\sigma(\hat{A}XW_1)W_2)$, we first approximate $\tilde{A}$ by $P'P^T$, then the whole GCN can be approximated as

$$\sigma(P'P^T\sigma(P'P^TXW_1)W_2) = \sigma(P'P^TP'\sigma(P^TXW_1)W_2) = P'\sigma(\sigma(P^TXW_1)W_2). \quad (11)$$

The first equality holds because $P'$ is a partition matrix and the last equality follows from the fact that $P^TP' = I$. Therefore, our analysis provides theoretical justifications of using StructComp as a substitute for non-linear deep GNNs.

## A.2 PROOF FOR THEOREM 4.1

**Single-view Graph Contrastive Learning.** We view adjacent nodes as positive pairs and non-adjacent nodes as negative pairs. In a coarse graph of supernodes, we define a pair of supernodes as positive if there exist two nodes within each supernode that are adjacent in the original graph. Let $f : \mathcal{X} \to \mathbb{R}^{d'}$ be a feature mapping. We are interested in a variant of the contrastive loss function that aims to minimize the distance between the features of positive pairs $v, v^+$.

$$\mathcal{L}(f) = \mathbb{E}_{v,v^+}\|f(v) - f(v^+)\|_2. \quad (12)$$

**Graph Partition.** For a graph $G = (V, E, X)$, we consider the Erdős-Rényi model, denoted as $G(n, p)$, where edges between $n$ vertices are included independently with probability $p$. We compute a partition $\mathcal{P}$ on the nodes. We denote $\mathcal{P} = \{S_1, \cdots, S_{n'}\}$, such that $V = \bigcup_{j=1}^{n'} S_j$ and $S_{j_1} \bigcap S_{j_2} = \emptyset$ for $j_1 \neq j_2$. We define a partition $\mathcal{P}$ to be an *even partition* if each subset has the same size $|S_1| = |S_2| = \cdots = |S_{n'}|$. For each vertex $v$, we denote $S(v) \in \mathcal{P}$ as the subset that $v$ belongs to. We denote $N(v) := \{s : (s, v) \in E \text{ or } (v, s) \in E\}$ as the set of neighbors of $v$. Denote $M(v) := N(v)\backslash S(v)$ be the set of $v$'s neighbors that are not in the subset with $v$.

For the partition $\mathcal{P}$, we denote $P' \in \{0, 1\}^{n \times n'}$ as its corresponding partition matrix, where $P'_{ij} = 1$ if the node $v_i$ belongs to the subset $S_j$. Let $R = A - P'P'^T$ be the partition remainder. By definition, $R_{ij} = 1$ if node $j \in M(i)$. Here we define the *partition loss* as

$$\mathcal{L}_{\text{partition}} = \|R\|_F := \|A - P'P'^T\|_F, \text{ where } \|\cdot\|_F \text{ denotes the Frobenius norm.}$$

Now we are ready to prove Theorem 4.1.

**Theorem 4.1.** *For the random graph $G(n, p)$ from Erdős-Rényi model, we construct an even partition $\mathcal{P} = \{S_1, \cdots, S_{n'}\}$. Let $f_G(X) = AXW$ be a feature mapping in the original graph and $f_{\mathcal{P}}(X) = P'^TXW$ as a linear mapping for the mixed nodes, where $W \in \mathbb{R}^{d \times d'}$. Then by conducting single-view contrastive learning, the contrastive loss for the original graph, denoted as $\mathcal{L}_G(W)$, can be approximated as the sum of the compressed contrastive loss, $\mathcal{L}_{\mathcal{P}}(W)$, and a term related to the low-rank approximation. Assume the features are bounded by $S_X := \max_i \|X_i\|_2$, we have*

$$|\mathcal{L}_G(W) - \mathcal{L}_{\mathcal{P}}(W)| \leq \|A - P'P'^T\|_F S_X\|W\|_2.$$

*Proof.* We denote the transpose of $i$-th row of $X$ as $X_i$. In the original graph, by the definition of adjacent matrix $A$, the embedding of each node $v_i$ is

$$f_{G,W}(v_i) = (AXW)_i = \sum_{v_s \in N(v_i)} W^TX_s.$$

For the compressed loss, each mixed node corresponds to a subset $S_i$ of the partition $\mathcal{P}$. We denote $B(S_i) := \{S_j \in \mathcal{P} : \exists u \in S_i, v \in S_j, s.t.(u, v) \in E\}$ as the set of mixed nodes that are connected to $S_i$. Then the embedding of mixed node $S_i$ can be written as

$$f_{\mathcal{P},W}(S_i) = (P'^TXW)_i$$

For a positive pair $u, v$ in the original graph, we can measure their difference in the feature subspace as follows.

$$
\begin{aligned}
f_{G,W}(u) - f_{G,W}(v) =& (AXW)_u - (AXW)_v \\
=& W^T \Big( \sum_{v_i \in S(u)} X_i - \sum_{v_i \in S(v)} X_i + \sum_{v_i \in M(u)} X_i - \sum_{v_i \in M(v)} X_i \Big) \\
=& (P'^T XW)_{S(u)} - (P'^T XW)_{S(v)} + (RXW)_u - (RXW)_v.
\end{aligned}
$$

For a positive pair $(S_i, S_j)$ in the compressed loss, their difference in the feature subspace is

$$
f_{\mathcal{P},W}(S_i) - f_{\mathcal{P},W}(S_j) = (P'^T XW)_i - (P'^T XW)_j.
$$

For any $i, j \in [n]$, $(v_i, v_j)$ is an edge in $G$ independently with probability $p$. We denote pos as the set of positive pairs in the compressed loss. Then by calculating contrastive loss on both graphs, we have

$$
\begin{aligned}
& \mathcal{L}_G(W) - \mathcal{L}_{\mathcal{P}}(W) \\
=& \mathop{\mathbb{E}}_{(u,v) \in E} \| f_{G,W}(u) - f_{G,W}(v) \|_2 - \mathop{\mathbb{E}}_{(S_i, S_j) \in \text{pos}} \| f_{\mathcal{P},W}(S_i) - f_{\mathcal{P},W}(S_j) \|_2 \\
=& \mathop{\mathbb{E}}_{(u,v) \in E} \| (P'^T XW)_{S(u)} - (P'^T XW)_{S(v)} + (RXW)_u - (RXW)_v \|_2 \\
& - \mathop{\mathbb{E}}_{(S_i, S_j) \in \text{pos}} \| (P'^T XW)_i - (P'^T XW)_j \|_2 \\
\leq& \mathop{\mathbb{E}}_{(u,v) \in E} \| (P'^T XW)_{S(u)} - (P'^T XW)_{S(v)} \|_2 - \mathop{\mathbb{E}}_{(S_i, S_j) \in \text{pos}} \| (P'^T XW)_i - (P'^T XW)_j \|_2 \\
& + \mathop{\mathbb{E}}_{(u,v) \in E} \| (RXW)_u - (RXW)_v \|_2 \\
=& \mathop{\mathbb{E}}_{(u,v) \in E} \| (RXW)_u - (RXW)_v \|_2.
\end{aligned}
$$

The last step holds because the partition is even. On the other hand,

$$
\begin{aligned}
& \mathcal{L}_{\mathcal{P}}(W) - \mathcal{L}_G(W) \\
=& \mathop{\mathbb{E}}_{(S_i, S_j) \in \text{pos}} \| (P'^T XW)_i - (P'^T XW)_j \|_2 \\
& - \mathop{\mathbb{E}}_{(u,v) \in E} \| (P'^T XW)_{D(u)} - (P'^T XW)_{D(v)} + (RXW)_u - (RXW)_v \|_2 \\
\leq& \mathop{\mathbb{E}}_{(S_i, S_j) \in \text{pos}} \| (P'^T XW)_i - (P'^T XW)_j \|_2 \\
& - \mathop{\mathbb{E}}_{(u,v) \in E} \| (P'^T XW)_{D(u)} - (P'^T XW)_{D(v)} \|_2 + \mathop{\mathbb{E}}_{(u,v) \in E} \| (RXW)_u - (RXW)_v \|_2 \\
=& \mathop{\mathbb{E}}_{(u,v) \in E} \| (RXW)_u - (RXW)_v \|_2.
\end{aligned}
$$

We combine both inequalities. Denote $\eta = \| A - P'P'^T \|_F$. Then we have

$$
\begin{aligned}
| \mathcal{L}_G(W) - \mathcal{L}_{\mathcal{P}}(W) | \leq& \mathop{\mathbb{E}}_{(u,v) \in E} \| (RXW)_u - (RXW)_v \|_2 \\
\leq& \| \delta^T XW \|_2 \qquad \triangleright \delta \in \mathbb{R}^n \text{ is an } \eta \text{ sparse vector with } 0, \pm 1 \text{ entries} \\
\leq& \sum_{i \in [n] : \delta_i \neq 0} \| X_i W \|_2 \\
\leq& \eta S_X \| W \|_2 \\
=& \| A - P'P'^T \|_F S_X \| W \|_2.
\end{aligned}
$$

$\square$

## A.3 AN UPPER BOUND OF THE APPROXIMATION GAP WITHOUT ER GRAPH

We give an extra analysis on arbitrary graphs. For non-random graphs, the approximation gap of losses is simply bounded by the Equation 4. Suppose the loss $\mathcal{L}$ is $L$-Lipschitz continuous,

$$|\mathcal{L}(P'P^T XW) - \mathcal{L}(\hat{A}^k XW)| \leq L \underbrace{\|P'P^T - \hat{A}^k\|}_{\text{Equation 4}} \|X\|\|W\|. \tag{13}$$

And for a spectral contrastive loss $\mathcal{L}_{\text{spec}}$, assume the graph partition are even, we have:

$$
\begin{aligned}
\mathcal{L}_{\text{spec}}(P^T XW) &= -\frac{2}{n}\sum_{i=1}^{n} e_{1,i}^T e_{2,i} + \frac{1}{n^2}\sum_{i=1}^{n}\sum_{j=1}^{n}(e_{1,i}^T e_{2,j})^2 \\
&= -\frac{2}{n}\sum_{k=1}^{n'}\sum_{i\in S_k} e_{1,i}^T e_{2,i} + \frac{1}{n^2}\sum_{i=1}^{n}\sum_{l=1}^{n'}\sum_{j\in S_l}(e_{1,i}^T e_{2,j})^2 \\
&= -\frac{2}{n'}\sum_{k=1}^{n'} E_{1,i}^T E_{2,i} + \frac{1}{nn'}\sum_{i=1}^{n}\sum_{l=1}^{n'}(e_{1,i}^T E_{2,j})^2 \\
&= -\frac{2}{n'}\sum_{k=1}^{n'} E_{1,i}^T E_{2,i} + \frac{1}{nn'}\sum_{k=1}^{n'}\sum_{i\in S_k}\sum_{l=1}^{n'}(e_{1,i}^T E_{2,j})^2 \\
&= -\frac{2}{n'}\sum_{k=1}^{n'} E_{1,i}^T E_{2,i} + \frac{1}{n'^2}\sum_{k=1}^{n'}\sum_{l=1}^{n'}(E_{1,i}^T E_{2,j})^2 = \mathcal{L}_{\text{spec}}(P'P^T XW),
\end{aligned}
\tag{14}
$$

where $e_{1,i}$ denotes the representations of a recovered node and $E_{1,i}^T$ denotes the representations of a compressed node. The above analysis shows that our approximation is reasonable for fixed graphs.

## A.4 PROOF FOR THEOREM 4.2

**Theorem 4.2.** *Consider a no-augmentation InfoNCE loss,*

$$\mathcal{L}_{\text{InfoNCE}} = \sum_i \sum_{j\in\text{pos}(i)} [h_i^T h_j] + \sum_i \sum_{j\in\text{neg}(i)} [\log(e^{h_i^T h_i} + e^{h_i^T h_j})]. \tag{15}$$

*Optimizing the expectation of this with augmentation $\mathbb{E}[\tilde{\mathcal{L}}_{\text{InfoNCE}}]$ introduce an additional regularization term, i.e.,*

$$\mathbb{E}[\tilde{\mathcal{L}}_{\text{InfoNCE}}] = \mathcal{L}_{\text{InfoNCE}} + \frac{1}{2}\sum_i \sum_{j\in\text{neg}(i)} \phi(h_i, h_j)\text{Var}(\tilde{h}_i), \tag{16}$$

*where $\phi(h_i, h_j) = \frac{(e^{h_i^2}h_i^2 + e^{h_i h_j}h_j^2)(e^{h_i^2} + e^{h_i h_j}) - (e^{h_i^2}h_i + e^{h_i h_j}h_j)^2}{2(e^{h_i^2} + e^{h_i h_j})^2}.$*

*Proof.*

$$\mathbb{E}[\tilde{\mathcal{L}}_{\text{InfoNCE}}] = \sum_i \sum_{j\in\text{pos}(i)} [h_i^T h_j] + \mathbb{E}[\Delta_1] + \sum_i \sum_{j\in\text{neg}(i)} [\log(e^{h_i^T h_i} + e^{h_i^T h_j})] + \mathbb{E}[\Delta_2], \tag{17}$$

where

$$\mathbb{E}[\Delta_1] = \mathbb{E}\left[\sum_i \sum_{j\in\text{pos}(i)} [h_j(\tilde{h}_i - h_i)]\right] = 0 \tag{18}$$

and

$$\mathbb{E}[\Delta_2] = \mathbb{E}\left[\sum_i \sum_{j \in \text{neg}} \log(e^{\tilde{h}_i h_i} + e^{\tilde{h}_i h_j}) - \log(e^{h_i h_i} + e^{h_i h_j})\right]$$

$$\approx \mathbb{E}[\sum_i \sum_{j \in \text{neg}} [\frac{e^{h_i h_i} h_i + e^{h_i h_j} h_j}{e^{h_i h_i} + e^{h_i h_j}}(\tilde{h}_i - h_i)$$

$$+ \frac{(e^{h_i^2} h_i^2 + e^{h_i h_j} h_j^2)(e^{h_i^2} + e^{h_i h_j}) - (e^{h_i^2} h_i + e^{h_i h_j} h_j)^2}{2(e^{h_i h_i} + e^{h_i h_j})^2}(\tilde{h}_i - h_i)^2]] \quad (19)$$

$$= \frac{1}{2}\sum_i \sum_{j \in \text{neg}} \frac{(e^{h_i^2} h_i^2 + e^{h_i h_j} h_j^2)(e^{h_i^2} + e^{h_i h_j}) - (e^{h_i^2} h_i + e^{h_i h_j} h_j)^2}{2(e^{h_i^2} + e^{h_i h_j})^2}\text{Var}(\tilde{h}_i).$$

Thus,

$$\mathbb{E}[\tilde{\mathcal{L}}_{\text{InfoNCE}}] = \mathcal{L}_{\text{InfoNCE}} + \frac{1}{2}\sum_i \sum_{j \in \text{neg}} \phi(h_i, h_j)\text{Var}(\tilde{h}_i), \quad (20)$$

where $\phi(h_i, h_j) = \frac{(e^{h_i^2} h_i^2 + e^{h_i h_j} h_j^2)(e^{h_i^2} + e^{h_i h_j}) - (e^{h_i^2} h_i + e^{h_i h_j} h_j)^2}{2(e^{h_i^2} + e^{h_i h_j})^2}.$ $\qquad\square$

## B    MORE RELATED WORK

**Graph contrastive learning.** Graph contrastive learning is an unsupervised representation learning technique that learns node representations by comparing similar and dissimilar sample pairs. Classical graph contrastive learning models, such as DGI (Veličković et al., 2018) and MVGRL (Hassani & Ahmadi, 2020), use a loss function based on mutual information estimation (Belghazi et al., 2018; Hjelm et al., 2019) to contrast node embeddings and graph embeddings. GRACE (Zhu et al., 2020) and its variants (Zhu et al., 2021b; You et al., 2020) strive to maximize the similarity of positive pairs while minimizing the similarity of negative pairs in augmented graphs, intending to learn more effective node embeddings. However, the computational complexity of these methods is too high for datasets, limiting their applications in large-scale graphs. To reduce computational consumption, CCA-SSG (Zhang et al., 2021b) simplified the loss function by eliminating negative pairs. Recently, GGD (Zheng et al., 2022) uses binary cross-entropy loss to distinguish between two groups of node samples, further reducing computational usage. Despite recent related work focusing on the scalability problem of graph contrastive learning (Wang et al., 2022), these methods still need to rely on graph sampling techniques when processing graphs with millions of nodes.

## C    IMPLEMENTATION DETAILS

The detailed statistics for the datasets used for transductive node classification are shown in Table 5. We compare GCL models trained with full graph and those trained with StructCompon small datasets. On Arxiv and Products, most GCL models cannot perform full graph training, so we compare the performance of different scalable training methods. For all GCL models, the learned representations are evaluated by training and testing a logistic regression classifier on smaller datasets. Due to Ogbn-Arxiv, Ogbn-Products and Ogbn-Papers100M exhibits more complex characteristics, we use a three-layers MLP as classifier.

Table 5: Summary of the datasets used in our experiments

| Dataset | Nodes | Edges | Features | Classes |
|---|---|---|---|---|
| Cora | 2,708 | 5,429 | 1,433 | 7 |
| Citeseer | 3,327 | 4,732 | 3,703 | 6 |
| Pubmed | 19,717 | 44,338 | 500 | 3 |
| Amazon-Photo | 7,650 | 238,163 | 745 | 8 |
| Amazon-Computers | 13,752 | 491,722 | 767 | 10 |
| Ogbn-Arxiv | 169,343 | 1,157,799 | 128 | 40 |
| Ogbn-Products | 2,449,029 | 61,859,140 | 100 | 47 |
| Papers100M | 111,059,956 | 1,615,685,872 | 128 | 172 |

**Details of Baselines.** We test the performance of StructComp on four GCL models: SCE[1], COLES[2],GRACE[3], CCA-SSG[4]. And we compared StructComp with three scalable training methods Cluster-GCN[5], Graphsaint[6], and Graphzoom[7] on large graphs.

**Details of our model.** In Table 6, we present the specific formulations of both the embeddings and the loss functions that we have trained in our experiments. All models are optimized using the Adam optimizer. The hyperparameters for GCL models trained with StructComp are basically the same as those used for full graph training of GCL models. We show the main hyperparameters in Table 7 and 8. The remaining hyperparameter settings for each GCL model are list in our code: `https://github.com/szzhang17/StructComp`.

Table 6: The compression and loss function of GCL models under StructComp framework.

| Model | Compression embedding | Loss function |
|---|---|---|
| SCE | $Z_c = X_c W$ | $\mathcal{L} = \frac{\alpha}{\text{Tr}(Z_c^T L_c^{neg} Z_c)}$ |
| COLES | $Z_c = X_c W$ | $\mathcal{L} = \text{Tr}(Z_c^T L_c^{neg} Z_c) - \text{Tr}(Z_c^T L_c Z_c)$ |
| GRACE | $\begin{cases} Z_c = \textbf{ReLU}((X_c W_1) W_2), \\ Z_c' = \textbf{ReLU}((X_c' W_1) W_2), \end{cases}$ | $\mathcal{L}(u,v) = \log \frac{e^{\phi(z_u, z_v)/\tau}}{e^{\phi(z_u, z_v)/\tau} + \sum_{k \neq u, k \in G_1} e^{\phi(z_u, z_k)/\tau} + \sum_{k \neq u, k \in G_2} e^{\phi(z_u, z_k)/\tau}}$ |
| CCA-SSG | $\begin{cases} Z_c = \textbf{ReLU}((X_c W_1) W_2), \\ Z_c' = \textbf{ReLU}((X_c' W_1) W_2), \end{cases}$ | $\mathcal{L} = \|\tilde{Z}_c - \tilde{Z}_c'\|_F^2 + \lambda(\|\tilde{Z}_c^T \tilde{Z}_c - I\|_F^2 + \|\tilde{Z}_c'^T \tilde{Z}_c' - I\|_F^2)$ |

Table 7: Summary of the main hyper-parameters on small datasets.

| Model | Cora | | Citeseer | | Pubmed | | Photo | | Computers | |
|---|---|---|---|---|---|---|---|---|---|---|
| | Lr | Epoch | Lr | Epoch | Lr | Epoch | Lr | Epoch | Lr | Epoch |
| SCE$_{\text{StructComp}}$ | 0.001 | 50 | 0.0001 | 50 | 0.02 | 25 | 0.001 | 20 | 0.001 | 20 |
| COLES$_{\text{StructComp}}$ | 0.001 | 20 | 0.0001 | 50 | 0.02 | 50 | 0.001 | 20 | 0.001 | 20 |
| GRACE$_{\text{StructComp}}$ | 0.001 | 20 | 0.0001 | 30 | 0.02 | 75 | 0.001 | 200 | 0.001 | 150 |
| CCA-SSG$_{\text{StructComp}}$ | 0.001 | 20 | 0.0001 | 50 | 0.01 | 50 | 0.002 | 100 | 0.005 | 40 |

**Configuration.** All the algorithms and models are implemented in Python and PyTorch Geometric. Experiments are conducted on a server with an NVIDIA 3090 GPU (24 GB memory) and an Intel(R) Xeon(R) Silver 4210R CPU @ 2.40GHz.

---

[1]SCE (MIT License): `https://github.com/szzhang17/Sparsest-Cut-Network-Embedding`

[2]COLES (MIT License): `https://github.com/allenhaozhu/COLES`

[3]GRACE (Apache License 2.0): `https://github.com/CRIPAC-DIG/GRACE`

[4]CCA-SSG (Apache License 2.0): `https://github.com/hengruizhang98/CCA-SSG`

[5]Cluster-GCN (MIT License): `https://github.com/pyg-team/pytorch_geometric/blob/master/examples/cluster_gcn_reddit.py`

[6]Graphsaint (MIT License): `https://github.com/pyg-team/pytorch_geometric/blob/master/examples/graph_saint.py`

[7]Graphzoom (MIT License): `https://github.com/cornell-zhang/GraphZoom`

Table 8: Summary of the main hyper-parameters on large datasets.

| | Ogbn-Arxiv | | | Ogbn-Products | | | Ogbn-Papers100M | | |
|---|---|---|---|---|---|---|---|---|---|
| | Lr | Epoch | Weight decay | Lr | Epoch | Weight decay | Lr | Epoch | Weight decay |
| SCE$_{StructComp}$ | 0.0001 | 10 | 0 | 0.0001 | 5 | 0.0005 | 0.001 | 1 | 0.0005 |
| COLES$_{StructComp}$ | 0.0001 | 5 | 0.0005 | 0.001 | 5 | 0.0005 | 0.001 | 1 | 0.0005 |
| GRACE$_{StructComp}$ | 0.001 | 5 | 0.0005 | 0.001 | 5 | 0.0005 | 0.001 | 1 | 0.0005 |
| CCA-SSG$_{StructComp}$ | 0.0001 | 10 | 0 | 0.001 | 10 | 0 | 0.001 | 1 | 0.0005 |

# D  MORE EXPERIMENTAL RESULTS AND DISCUSSIONS

## D.1  EXPERIMENTS ON OGBN-PAPERS100M

We conduct experiments on the Ogbn-Papers100M dataset. The experimental results are shown in Table 9. We use StructComp to train four representative GCL models. Here, we compressed ogbn-papers100M into a feature matrix $X_c \in R^{5000*128}$ and trained GCL using StructComp. The table also presents the results of GGD trained with ClusterGCN. Although GGD is specifically designed for training large graphs, when dealing with datasets of the scale of ogbn-papers100M, it still requires graph sampling to construct subgraphs and train GGD on a large number of subgraphs. In contrast, our StructComp only requires training a simple and small-scale MLP, resulting in significantly lower resource consumption compared to GGD+ClusterGCN.

Table 9: The accuracy, training time per epoch and memory usage on the Ogbn-Papers100M dataset.

| Method | Acc | Time | Mem |
|---|---|---|---|
| GGD | 63.5±0.5 | 1.6h | 4.3GB |
| SCE$_{StructComp}$ | 63.6±0.4 | 0.18s | 0.1GB |
| COLES$_{StructComp}$ | 63.6±0.4 | 0.16s | 0.3GB |
| GRACE$_{StructComp}$ | 64.0±0.3 | 0.44s | 0.9GB |
| CCA-SSG$_{StructComp}$ | 63.5±0.2 | 0.18s | 0.1GB |

## D.2  EXPERIMENTS ON THE STABILITY OF STRUCTCOMP

We conduct extra experiments to study the robustness of StructComp. We randomly add 10% of noisy edges into three datasets and perform the node classification task. The experimental results are shown in Table 10. On the original dataset, the models trained with StructComp showed performance improvements of 0.36, 0.40, 1.30 and 1.87, respectively, compared to the models trained with full graphs. With noisy perturbation, the models trained with StructComp showed performance improvements of 0.80, 1.27, 2.47, and 1.87, respectively, compared to full graph training. This indicates that GCL models trained with StructComp exhibit better robustness.

Table 10: The results over 50 random splits on the perturbed datasets.

| Method | Cora | Citeseer | Pubmed |
|---|---|---|---|
| SCE | 78.8±1.2 | 69.7±1.0 | 73.4±2.2 |
| SCE$_{StructComp}$ | 79.3±0.9 | 69.3±0.9 | 75.7±2.8 |
| COLES | 78.7±1.2 | 68.0±1.0 | 66.5±1.8 |
| COLES$_{StructComp}$ | 79.0±1.0 | 68.3±0.9 | 69.7±2.6 |
| GRACE | 77.6±1.1 | 64.1±1.4 | 64.5±1.7 |
| GRACE$_{StructComp}$ | 78.3±0.8 | 69.1±0.9 | 66.2±2.4 |
| CCA-SSG | 75.5±1.3 | 69.1±1.2 | 73.5±2.2 |
| CCA-SSG$_{StructComp}$ | 78.2±0.7 | 69.2±0.8 | 76.3±2.5 |

## D.3 Experiments on deep GNN encoder

In order to verify the approximation quality to the diffusion matrix of StructComp, we test the performance on a deep GNN architecture called SSGC (Zhu & Koniusz, 2021). We transferred the trained parameters of StructComp to the SSGC encoder for inference. For full graph training in GCL, both the training and inference stages were performed using the SSGC encoder. Table 11 shows our experimental results, indicating that even with a deeper and more complicated encoder, StructComp still achieved outstanding performance.

Table 11: The results of GCLs with SSGC encoders over 50 random splits.

| Method | Cora | Citeseer | Pubmed |
|---|---|---|---|
| SCE | 81.8±0.9 | 72.0±0.9 | 78.4±2.8 |
| SCE$_{StructComp}$ | 82.0±0.8 | 71.7±0.9 | 77.8±2.9 |
| COLES | 81.8±0.9 | 71.3±1.1 | 74.8±3.4 |
| COLES$_{StructComp}$ | 82.0±0.8 | 71.6±1.0 | 75.6±3.0 |
| GRACE | 80.2±0.8 | 70.7±1.0 | 77.3±2.7 |
| GRACE$_{StructComp}$ | 81.1±0.8 | 71.0±1.0 | 78.2±1.3 |
| CCA-SSG | 82.1±0.9 | 71.9±0.9 | 78.2±2.8 |
| CCA-SSG$_{StructComp}$ | 82.6±0.7 | 71.7±0.9 | 79.4±2.6 |

## D.4 Comparison with recent GCL baselines

We provide a comparison between StructComp and recent GCL baselines (Li et al., 2023; Wang et al., 2023; Ma et al., 2023; Zheng et al., 2022). The specific results are shown in Table 12. For SP-GCL, we are unable to get the classification accuracy on CiteSeer since it does not take isolated nodes as input. The performance and resource consumption of various GCL models trained with StructComp are superior to recent GCL baselines.

It should be noted that the goal of these studies and our work are different. The aim of SPGCL is to handle homophilic graphs and heterophilic graphs simultaneously. BlockGCL attempts to explore the application of deep GNN encoder in the GCL field. Contrast-Reg is a novel regularization method which is motivated by the analysis of expected calibration error. GGD is a GCL model specifically designed for training large graphs, it is not a training framework. On the other hand, StructComp is a framework designed to scale up the training of GCL models: it aims to efficiently train common GCL models without performance drop. It is not a new GCL model that aims to achieve SOTA performance compared to existing GCL models. So our work is orthogonal to these previous works. In fact, StructComp can be used as the training method for SP-GCL, BlockGCL, Contrast-Reg and GGD. In future work, we will further investigate how to train these recent graph contrastive learning methods using StructComp.

Table 12: The results of StructComp-trained GCLs and some GCL baselines over 50 random splits.

| Method | Cora | | | Citeseer | | | Pubmed | | |
|---|---|---|---|---|---|---|---|---|---|
| | Acc | Time | Mem | Acc | Time | Mem | Acc | Time | Mem |
| BlockGCL | 78.1±2.0 | 0.026 | 180 | 64.5±2.0 | 0.023 | 329 | 74.7±3.1 | 0.037 | 986 |
| SP-GCL | 81.4±1.2 | 0.016 | 247 | - | 0.021 | 319 | 74.8±3.2 | 0.041 | 1420 |
| Contrast-Reg | 79.2±1.3 | 0.048 | 355 | 69.8±1.6 | 0.097 | 602 | 72.4±3.5 | 0.334 | 11655 |
| GGD | 79.9±1.7 | 0.013 | 118 | 71.3±0.7 | 0.018 | 281 | 74.0±2.4 | 0.015 | 311 |
| SCE$_{StructComp}$ | 81.6±0.9 | 0.002 | 23 | 71.5±1.0 | 0.002 | 59 | 77.2±2.9 | 0.003 | 54 |
| COLES$_{StructComp}$ | 81.8±0.8 | 0.002 | 24 | 71.6±0.9 | 0.003 | 60 | 75.3±3.1 | 0.003 | 61 |
| GRACE$_{StructComp}$ | 79.7±0.9 | 0.009 | 37 | 70.5±1.0 | 0.009 | 72 | 77.2±1.4 | 0.009 | 194 |
| CCA-SSG$_{StructComp}$ | 82.3±0.8 | 0.006 | 38 | 71.6±0.9 | 0.005 | 71 | 78.3±2.5 | 0.006 | 85 |

## D.5 Discussion on Graph Partitioning

We conducted additional experiments to investigate the impact of graph partitioning on the performance of StructComp. In Table 13, we demonstrate the effects of three algorithms, algebraic JC,

variation neighborhoods, and affinity GS, on the performance of StructComp. These three graph coarsening algorithms are widely used in scalable GNNs (Huang et al., 2021), from which we can obtain the specific graph partition matrix P. The experimental results suggest that different graph partition methods has little impact on StructComp on these datasets.

Table 13: The results of different graph partition methods.

| Method | Cora | Citeseer | Pubmed |
|---|---|---|---|
| VN+SCE$_{StructComp}$ | 81.3±0.8 | 71.5±1.0 | 77.5±2.7 |
| JC+SCE$_{StructComp}$ | 81.2±0.9 | 71.5±1.1 | 77.3±2.7 |
| GS+SCE$_{StructComp}$ | 81.5±0.8 | 71.4±1.0 | 77.4±3.0 |
| METIS+SCE$_{StructComp}$ | 81.6±0.9 | 71.5±1.0 | 77.2±2.9 |
| VN+COLES$_{StructComp}$ | 81.4±0.9 | 71.6±0.9 | 75.5±3.0 |
| JC+COLES$_{StructComp}$ | 81.4±0.9 | 71.5±1.0 | 75.3±3.0 |
| GS+COLES$_{StructComp}$ | 81.8±0.8 | 71.6±1.0 | 75.5±3.2 |
| METIS+COLES$_{StructComp}$ | 81.8±0.8 | 71.6±0.9 | 75.3±3.1 |

## D.6 EXPERIMENTS ON INDUCTIVE DATASETS

Our StructComp can also be used to handle inductive node classification tasks (Hamilton et al., 2017; Zeng et al., 2021). We provide additional experiments on inductive node classification in Table 14. Clearly, the GCL models trained with StructComp also perform exceptionally well on inductive node classification tasks.

Table 14: The results on two inductive datasets. OOM means Out of Memory on GPU.

| Method | Flickr | | | Reddit | | |
|---|---|---|---|---|---|---|
| | Acc | Time | Mem | Acc | Time | Mem |
| SCE | 50.6 | 0.55 | 8427 | - | - | OOM |
| SCE$_{StructComp}$ | 51.6 | 0.003 | 43 | 94.4 | 0.017 | 1068 |
| COLES | 50.3 | 0.83 | 9270 | - | - | OOM |
| COLES$_{StructComp}$ | 50.7 | 0.003 | 48 | 94.2 | 0.024 | 1175 |
| GRACE | - | - | OOM | - | - | OOM |
| GRACE$_{StructComp}$ | 51.5 | 0.010 | 221 | 94.3 | 0.079 | 8683 |
| CCA-SSG | 51.6 | 0.125 | 1672 | 94.9 | 0.21 | 5157 |
| CCA-SSG$_{StructComp}$ | 51.8 | 0.007 | 99 | 95.2 | 0.56 | 457 |

## D.7 EXPERIMENTS ON HETEROPHILOUS GRAPHS

We conduct experiments to train SP-GCL (Wang et al., 2023) with StructComp, in order to verify the performance of StructComp on heterophilous graphs. The experimental results are shown in Table 15. Overall, the SP-GCL trained by StructComp is superior to full graph training. This is our initial attempt to use StructComp to handle heterophilous graphs, and it is obviously a valuable direction worth further research.

Table 15: The results on heterophilous datasets.

| Method | Chameleon | | | Squirrel | | | Actor | | |
|---|---|---|---|---|---|---|---|---|---|
| | Acc | Time | Mem | Acc | Time | Mem | Acc | Time | Mem |
| SP-GCL | 65.28±0.53 | 0.038 | 739 | 52.10±0.67 | 0.080 | 3623 | 28.94±0.69 | 0.041 | 802 |
| SP-GCL$_{StructComp}$ | 66.65±1.63 | 0.011 | 168 | 53.08±1.39 | 0.009 | 217 | 28.70±1.25 | 0.013 | 159 |

