# OpenReview forum: "StructComp: Substituting propagation with Structural Compression in Training Graph Contrastive Learning"
_ICLR.cc/2024/Conference — ICLR 2024 poster_

### Official Review · Reviewer_kF3D · 2023-10-30

**Soundness:** 3 good
**Presentation:** 3 good
**Contribution:** 3 good
**Rating:** 6
**Confidence:** 4

**Summary:**

This paper studies improving efficiency of graph contrastive learning. The authors propose a structural compression framework, StructComp, that adopts a low-rank approximation of the diffusion matrix to obtain compressed node embeddings. They show that the original GCL loss can be approximated with the contrastive loss computed by StructComp, with an additional benefit of the robustness. Experiments on seven benchmark datasets show that StructComp greatly reduces the time and memory consumption while improving model performance compared to the vanilla GCL models and scalable training methods.

**Strengths:**

(+) The proposed structural compression idea is new and interesting;

(+) The presentation and organization are clear and easy to follow;

**Weaknesses:**

(-) The applicability of StructComp seems to be limited;

(-) Several claims have not been verified;

(-) Some related works have not been compared or discussed;

**Questions:**

1. The applicability of StructComp seems to be limited:

- Theoretically, StructComp has to rely on the approximation of the diffusion matrix for a specific graph and GNN model, as demonstrated in Theorem 4.1. How well StructComp can approximate to more complicated graphs such as heterophilous graphs, and more complicated while commonly used GNNs such as GraphSage, GIN, GAT, or even more interesting variants such as PNA?

- Empirically, what is the exact setting of StructComp for node classification? Can StructComp be applied to both transductive and inductive node classification?

- How well can StructComp approximate different augmentations in GCL?

2. Several claims have not been verified:

- The paper claims that StructComp can work for large scale graphs, while the benchmarked datasets are rather small or medium scale. Although it’s claimed that ogbn-products and arxiv are large datasets, while they are indeed medium scale datasets according to OGB (https://ogb.stanford.edu/docs/nodeprop/). To support the claim, it’s expected to evaluate StructComp in large datasets such as papers100M, reddit, or OGBG-LSC datasets.

- The paper also claims that StructComp has better robustness and stability with the additional regularization, while no evidence’s been found.

3. Some related works have not been compared or discussed:

-  Some GCL works have not been discussed in the paper, for example, [1,2,3].

- Why not comparing efficient GCL baselines such as CCA-SSG, and GGD discussed in the paper?

4. How the time and memory cost are computed? Do they count in the preprocessing steps?


**References**

[1] Calibrating and Improving Graph Contrastive Learning, TMLR’23.

[2] Single-Pass Contrastive Learning Can Work for Both Homophilic and Heterophilic Graph, TMLR’23.

[3] Scaling Up, Scaling Deep: Blockwise Graph Contrastive Learning, arXiv’23.

[4] Structure-free Graph Condensation: From Large-scale Graphs to Condensed Graph-free Data, arXiv’23.

---

> ### Author Response · Authors · 2023-11-16
> **Response to Reviewer kF3D[1/4]**
>
> Thank you for your detailed review. We would like to address your questions/concerns below:
>
> **Q1:Theoretically, StructComp has to rely on the approximation of the diffusion matrix for a specific graph and GNN model, as demonstrated in Theorem 4.1. How well StructComp can approximate to more complicated graphs such as heterophilous graphs, and more complicated while commonly used GNNs such as GraphSage, GIN, GAT, or even more interesting variants such as PNA?**
>
>
>
> In order to verify the approximation quality to the diffusion matrix of StructComp, we test the performance on a deep GNN architecture called SSGC [1]. We transferred the trained parameters of StructComp to the SSGC encoder for inference. For full graph training in GCL, both the training and inference stages were performed using the SSGC encoder. Table 1 shows our experimental results, indicating that even with a deeper and more complicated encoder, StructComp still achieved outstanding performance.
>
>
>
> Table 1: The results of GCLs with SSGC encoders over 50 random splits.
>
> | Method                        | Cora         | Citeseer     | Pubmed       |
> | ----------------------------- | ------------ | ------------ | ------------ |
> | SCE                           | 81.8$\pm$0.9 | 72.0$\pm$0.9 | 78.4$\pm$2.8 |
> | SCE$_{\text{StructComp}}$     | 82.0$\pm$0.8 | 71.7$\pm$0.9 | 77.8$\pm$2.9 |
> | COLES                         | 81.8$\pm$0.9 | 71.3$\pm$1.1 | 74.8$\pm$3.4 |
> | COLES$_{\text{StructComp}}$   | 82.0$\pm$0.8 | 71.6$\pm$1.0 | 75.6$\pm$3.0 |
> | GRACE                         | 80.2$\pm$0.8 | 70.7$\pm$1.0 | 77.3$\pm$2.7 |
> | GRACE$_{\text{StructComp}}$   | 81.1$\pm$0.8 | 71.0$\pm$1.0 | 78.2$\pm$1.3 |
> | CCA-SSG                       | 82.1$\pm$0.9 | 71.9$\pm$0.9 | 78.2$\pm$2.8 |
> | CCA-SSG$_{\text{StructComp}}$ | 82.6$\pm$0.7 | 71.7$\pm$0.9 | 79.4$\pm$2.6 |
>
>
>
> To the best of our knowledge, current mainstream GCL models, whether single-view models such as SCE, COLES, GLEN [2], or multi-view models such as DGI, GRACE, CCA-SSG, GGD, and their variants, all use SGC or GCN encoders. These unsupervised GCL models can achieve performance surpassing supervised GNNs (such as GAT, SAGE) with simple encoders on many datasets. Few GCL models have focused on whether using different encoders can achieve better performance. Moreover, GCL training is more complicated than supervised GNNs, so introducing complex encoders may make the models more difficult to train. Considering this, we did not extensively explore how using other encoders would affect StructComp. We believe this is a good question, and we will investigate it further in future work.
>
> On the other hand, we have not implemented StructComp on heterophilous graphs , as the chosen basic GCL methods - SCE, COLES, GRACE, and CCA-SSG - are not suitable for heterophilous graph. In our future work, we will explore applying StructComp to more complex graph data structures.
>
>
>
> **Q2:Empirically, what is the exact setting of StructComp for node classification? Can StructComp be applied to both transductive and inductive node classification?**
>
> Our StructComp can also be used to handle inductive node classification tasks. We provide additional experiments on inductive node classification in Table 2. Clearly, the GCL models trained with StructComp also perform exceptionally well on inductive node classification tasks.
>
> Table 2: The results on two inductive datasets. OOM means Out of Memory on GPU.
>
> | Method                        | Flickr Acc | Flickr Time | Flickr Mem | Reddit Acc | Reddit Time | Reddit Mem |
> | ----------------------------- | ---------- | ----------- | ---------- | ---------- | ----------- | ---------- |
> | SCE                           | 50.6       | 0.55        | 8427       | -          | -           | OOM        |
> | SCE$_{\text{StructComp}}$     | 51.6       | 0.003       | 43         | 94.4       | 0.017       | 1068       |
> | COLES                         | 50.3       | 0.83        | 9270       | -          | -           | OOM        |
> | COLES$_{\text{StructComp}}$   | 50.7       | 0.003       | 48         | 94.2       | 0.024       | 1175       |
> | GRACE                         | -          | -           | OOM        | -          | -           | OOM        |
> | GRACE$_{\text{StructComp}}$   | 51.5       | 0.010       | 221        | 94.3       | 0.079       | 8683       |
> | CCA-SSG                       | 51.6       | 0.125       | 1672       | 94.9       | 0.21        | 5157       |
> | CCA-SSG$_{\text{StructComp}}$ | 51.8       | 0.007       | 99         | 95.2       | 0.56        | 457        |

---

> ### Author Response · Authors · 2023-11-16
> **Response to Reviewer kF3D[2/4]**
>
> **Q3: How well can StructComp approximate different augmentations in GCL?**
>
> For multi-view GCLs, we designed a new data augmentation method, Dropmember, for our StructComp. It is important to note that Dropmember is not designed to approximate other graph data augmentation methods, but rather because previous graph data augmentation methods cannot be used under the StructComp framework. According to the analysis in DropMessage [3], common graph data augmentations can be unified under a single framework, and we have proved that our Dropmember falls within this framework as well.
>
> **Q4:The paper claims that StructComp can work for large scale graphs, while the benchmarked datasets are rather small or medium scale. Although it’s claimed that ogbn-products and arxiv are large datasets, while they are indeed medium scale datasets according to OGB. To support the claim, it’s expected to evaluate StructComp in large datasets such as papers100M, reddit, or OGBG-LSC datasets.**
>
> We have conducted extra experiments on the ogbn-papers100M dataset according to your suggestion. We use StructComp to train four representative GCL models. Here, we compressed ogbn-papers100M into a feature matrix $X_c \in R^{5000*128}$ and trained GCL using StructComp. The table also presents the results of GGD trained with ClusterGCN. Although GGD is specifically designed for training large graphs, when dealing with datasets of the scale of ogbn-papers100M, it still requires graph sampling to construct subgraphs and train GGD on a large number of subgraphs. In contrast, our StructComp only requires training a simple and small-scale MLP, resulting in significantly lower resource consumption compared to GGD+ClusterGCN.
>
>
>
> Table 3: The accuracy, training time per epoch and memory usage on the Ogbn-papers100M dataset.
>
> | Method                        | Acc          | Time  | Mem   |
> | ----------------------------- | ------------ | ----- | ----- |
> | GGD                           | 63.5$\pm$0.5 | 1.6h  | 4.3GB |
> | SCE$_{\text{StructComp}}$     | 63.6$\pm$0.4 | 0.18s | 0.1GB |
> | COLES$_{\text{StructComp}}$   | 63.6$\pm$0.4 | 0.16s | 0.3GB |
> | GRACE$_{\text{StructComp}}$   | 64.0$\pm$0.3 | 0.44s | 0.9GB |
> | CCA-SSG$_{\text{StructComp}}$ | 63.5$\pm$0.2 | 0.18s | 0.1GB |
>
>
>
> **Q5:The paper also claims that StructComp has better robustness and stability with the additional regularization, while no evidence’s been found.**
>
> We have conducted extra experiments to study the robustness of StructComp. We randomly add 10\% of noisy edges into three datasets and perform the node classification task. On the original dataset, the models trained with StructComp showed performance improvements of **0.36**, **0.40**, **1.30** and **1.87**, respectively, compared to the models trained with full graphs. With noisy perturbation, the models trained with StructComp showed performance improvements of **0.80**, **1.27**, **2.47**, and **1.87**, respectively, compared to full graph training. This indicates that GCL models trained with StructComp exhibit better robustness.
>
>
>
> Table 4: The results over 50 random splits  on the perturbed datasets.
>
> | Method                        | Cora         | CiteSeer     | PubMed       |
> | ----------------------------- | ------------ | ------------ | ------------ |
> | SCE                           | 78.8$\pm$1.2 | 69.7$\pm$1.0 | 73.4$\pm$2.2 |
> | SCE$_{\text{StructComp}}$     | 79.3$\pm$0.9 | 69.3$\pm$0.9 | 75.7$\pm$2.8 |
> | COLES                         | 78.7$\pm$1.2 | 68.0$\pm$1.0 | 66.5$\pm$1.8 |
> | COLES$_{\text{StructComp}}$   | 79.0$\pm$1.0 | 68.3$\pm$0.9 | 69.7$\pm$2.6 |
> | GRACE                         | 77.6$\pm$1.1 | 64.1$\pm$1.4 | 64.5$\pm$1.7 |
> | GRACE$_{\text{StructComp}}$   | 78.3$\pm$0.8 | 69.1$\pm$0.9 | 66.2$\pm$2.4 |
> | CCA-SSG                       | 75.5$\pm$1.3 | 69.1$\pm$1.2 | 73.5$\pm$2.2 |
> | CCA-SSG$_{\text{StructComp}}$ | 78.2$\pm$0.7 | 69.2$\pm$0.8 | 76.3$\pm$2.5 |

---

> ### Author Response · Authors · 2023-11-16
> **Response to Reviewer kF3D[3/4]**
>
> **Q6:Some GCL works have not been discussed in the paper, for example, [1,2,3].**
>
> It should be noted that the goal of these studies and our work are different. The aim of SPGCL is to handle homophilic graphs and heterophilic graphs simultaneously. BlockGCL attempts to explore the application of deep GNN encoder in the GCL field. Contrast-Reg is a novel regularization method which is motivated by the analysis of expected calibration error. On the other hand, StructComp is a framework designed to scale up the training of GCL models: it aims to efficiently train common GCL models without performance drop. It is not a new GCL model that aims to achieve  SOTA performance compared to existing GCL models. So our work is orthogonal to these three previous works. In fact, StructComp can be used as the training method for SP-GCL, BlockGCL and Contrast-Reg. In future work, we will further investigate how to train these recent graph contrastive learning methods using StructComp.
>
>
>
> In terms of scalability, which is the main goal of our work, we have conducted extra experiments to compare SP-GCL, BlockGCL, Contrast-Reg and StructComp-trained baseline. The results confirm that the aforementioned three GCL models are not designed for reducing training costs.
>
>
>
> Table 5: The results of StructComp-trained GCLs and some GCL baselines over 50 random splits. For SP-GCL, we are unable to get the classification accuracy on CiteSeer since it does not take isolated nodes as input.
>
> | Method                                                       | Cora Acc     | Cora Time | Cora Mem | CiteSeer Acc | CiteSeer Time | CiteSeer Mem | PubMed Acc   | PubMed Time | PubMed Mem |
> | ------------------------------------------------------------ | ------------ | --------- | -------- | ------------ | ------------- | ------------ | ------------ | ----------- | ---------- |
> | BlockGCL                                                     | 78.1$\pm$2.0 | 0.026     | 180      | 64.5$\pm$2.0 | 0.023         | 329          | 74.7$\pm$3.1 | 0.037       | 986        |
> | SP-GCL                                                       | 81.4$\pm$1.2 | 0.016     | 247      | -            | 0.021         | 319          | 74.8$\pm$3.2 | 0.041       | 1420       |
> | Contrast-Reg                                                 | 79.2$\pm$1.3 | 0.048     | 355      | 69.8$\pm$1.6 | 0.097         | 602          | 72.4$\pm$3.5 | 0.334       | 11655      |
> | SCE$_{\text{StructComp}}$                                    | 81.6$\pm$0.9 | 0.002     | 23       | 71.5$\pm$1.0 | 0.002         | 59           | 77.2$\pm$2.9 | 0.003       | 54         |
> | COLES$_{\text{StructComp}}$                                  | 81.8$\pm$0.8 | 0.002     | 24       | 71.6$\pm$0.9 | 0.003         | 60           | 75.3$\pm$3.1 | 0.003       | 61         |
> | GRACE$_{\text{StructComp}}$                                  | 79.7$\pm$0.9 | 0.009     | 37       | 70.5$\pm$1.0 | 0.009         | 72           | 77.2$\pm$1.4 | 0.009       | 194        |
> | CCA-SSG$_{\text{StructComp}}$ &nbsp;&nbsp;&nbsp;&nbsp;&nbsp;&nbsp; &nbsp; | 82.3$\pm$0.8 | 0.006     | 38       | 71.6$\pm$0.9 | 0.005         | 71           | 78.3$\pm$2.5 | 0.006       | 85         |

---

> ### Author Response · Authors · 2023-11-16
> **Response to Reviewer kF3D[4/4]**
>
> **Q7:Why not comparing efficient GCL baselines such as CCA-SSG, and GGD discussed in the paper?**
>
> *We have compared the performance of CCA-SSG with CCA-SSG trained using StructComp in the experimental section of the original submission. The specific results can be found in table 1 and table 2 of section 6.* Clearly, the performance of CCA-SSG trained with StructComp far surpasses the original CCA-SSG. Moreover, we add a comparison between StructComp and GGD in the new submission, the specific results are shown in the table below. The performance and resource consumption of various GCL models trained with StructComp are superior to GGD.
>
> Table 6: The results of StructComp-trained GCLs and GGD.
>
> | Method                        | Cora Acc     | Cora Time | Cora Mem | CiteSeer Acc | CiteSeer Time | CiteSeer Mem | PubMed Acc   | PubMed Time | PubMed Mem |
> | ----------------------------- | ------------ | --------- | -------- | ------------ | ------------- | ------------ | ------------ | ----------- | ---------- |
> | GGD                           | 79.9$\pm$1.7 | 0.013     | 118      | 71.3$\pm$0.7 | 0.018         | 281          | 74.0$\pm$2.4 | 0.015       | 311        |
> | SCE$_{\text{StructComp}}$     | 81.6$\pm$0.9 | 0.002     | 23       | 71.5$\pm$1.0 | 0.002         | 59           | 77.2$\pm$2.9 | 0.003       | 54         |
> | COLES$_{\text{StructComp}}$   | 81.8$\pm$0.8 | 0.002     | 24       | 71.6$\pm$0.9 | 0.003         | 60           | 75.3$\pm$3.1 | 0.003       | 61         |
> | GRACE$_{\text{StructComp}}$   | 79.7$\pm$0.9 | 0.009     | 37       | 70.5$\pm$1.0 | 0.009         | 72           | 77.2$\pm$1.4 | 0.009       | 194        |
> | CCA-SSG$_{\text{StructComp}}$ | 82.3$\pm$0.8 | 0.006     | 38       | 71.6$\pm$0.9 | 0.005         | 71           | 78.3$\pm$2.5 | 0.006       | 85         |
>
>
>
> **Q8:How the time and memory cost are computed? Do they count in the preprocessing steps?**
>
> We compute the time and memory cost in the same way as previous work on scalable GNNs: the time is the training time per epoch and memory is the peak GPU memory cost during training. The preprocessing time for our StructComp is strictly less than common scalable methods (e.g., ClusterGCN, GraphSaint and GraphAutoScale), since it only performs METIS once (like ClusterGCN) and does not need to do random sampling in each epoch. METIS is highly scalable and even papers100M (with 100M nodes) can be processed within an hour on a commercial CPU. Thus for big graphs, the prepocessing time is dominated by the training time, and more importantly, it only needs to be done once and the partition result can be shared by all experiments.
>
> ------------------------------------------------------
>
> The discussion of all the aforementioned issues will be added into the revised version of our paper. We appreciate your insightful feedback once again.
>
> [1]H. Zhu, and P. Koniusz. Simple spectral graph convolution. ICLR 2020.
>
> [2]H. Zhu, and P. Koniusz. Generalized Laplacian Eigenmaps. Neurips 2022.
>
> [3]Fang, T., Xiao, Z., Wang, C., Xu, J., Yang, X., and Yang, Y. Dropmessage: Unifying random dropping for graph neural networks. AAAI 2023.

---

> ### Author Response · Authors · 2023-11-20
> **A friendly reminder for discussion and we appreciate your input**
>
> Dear Reviewer kF3D,
>
> We thank you again for your insightful and constructive review. We have worked hard and have thoroughly addressed your comments in the rebuttal.
>
> As the discussion period soon comes to an end, we are looking forward to your feedback to our response and revised manuscript. Many thanks again for your time and efforts.
>
> Best regards,
>
> Authors of Submission 5510

---

> > ### Comment · Reviewer_kF3D · 2023-11-20
> > **Thank you for the reply**
> >
> > Thank you for the extensive additional experiments and the comprehensive discussion. Most of my concerns are resolved, nevertheless, there remain some questions regarding the response:
> >
> > - Regarding Q1: Since the heterophilous graphs have received lots of attention from the community, it'd be better if StructComp could be implemented for both kinds of graphs, with the suitable GCL method such as SP-GCL.
> > - Regarding Q8: In practice, the pre-processing time must be considered to find the best trade-off in terms of performance and efficiency, considering different scales of the graphs (the small, medium scale graphs in the original submission; and the large-scale graphs presented in the authors' response). It's important to present the efficiency results including the full pipeline of different methods, i.e., the overall time/mem cost for training the model, and the overhead for inference.

---

> ### Author Response · Authors · 2023-11-21
> **Further Response to Reviewer kF3D[1/2]**
>
> Thank you for your feedback . We would like to address your questions/concerns below:
>
> **Regarding Q1: Since the heterophilous graphs have received lots of attention from the community, it'd be better if StructComp could be implemented for both kinds of graphs, with the suitable GCL method such as SP-GCL.**
>
> We provide experiments of training SP-GCL with StructComp to verify the performance of StructComp on heterophilous graphs. The experimental results are shown in Table 7. Overall, the SP-GCL trained by StructComp is superior to full graph training. This is our initial attempt to use StructComp to handle heterophilous graphs, and it is obviously a valuable direction worth further research.
>
> Table 7. The results on heterophilous datasets.
>
> |                              | Chameleon Acc  | Chameleon Time | Chameleon Mem | Squirrel Acc   | Squirrel Time | Squirrel Mem | Actor Acc      | Actor Time | Actor Mem |
> | ---------------------------- | -------------- | -------------- | ------------- | -------------- | ------------- | ------------ | -------------- | ---------- | --------- |
> | SP-GCL                       | 65.28$\pm$0.53 | 0.038          | 739           | 52.10$\pm$0.67 | 0.080         | 3623         | 28.94$\pm$0.69 | 0.041      | 802       |
> | SP-GCL$_{\text{StructComp}}$ | 66.65$\pm$1.63 | 0.011          | 168           | 53.08$\pm$1.39 | 0.009         | 217          | 28.70$\pm$1.25 | 0.013      | 159       |
>
> **Regarding Q8: In practice, the pre-processing time must be considered to find the best trade-off in terms of performance and efficiency, considering different scales of the graphs (the small, medium scale graphs in the original submission; and the large-scale graphs presented in the authors' response). It's important to present the efficiency results including the full pipeline of different methods, i.e., the overall time/mem cost for training the model, and the overhead for inference.**
>
> As we said in our previous response, the pre-processing of our StructComp only needs to perform METIS once. The specific time for METIS on our CPU is shown in Table 8.
>
> Table 8. Preprocessing time.
>
> |       | Cora   | Citeseer | Pubmed | Computers | Photo | Flickr | Reddit | Arxiv | Products | 100M |
> | ----- | ------ | -------- | ------ | --------- | ----- | ------ | ------ | ----- | -------- | ---- |
> | METIS | 0.075s | 0.059s   | 0.60s  | 1.4s      | 0.58s | 1.7s   | 26.9s  | 15.7s | 225s     | 1.4h |
>
> For small and medium datasets, the time required by METIS is very short. Therefore, in these basic GCL models, the overall time required for full graph training is still more than that of StructComp. The results are shown in Table 9.
>
> Table 9. The overall time (seconds) and memory usage (MB) on small and medium datasets.
>
> |                               | Cora Mem | Cora Time | Citeseer Mem | Citeseer Time | Pubmed Mem | Pubmed Time | Computers Mem | Computers Time | Photo Mem | Photo Time |
> | ----------------------------- | -------- | --------- | ------------ | ------------- | ---------- | ----------- | ------------- | -------------- | --------- | ---------- |
> | SCE                           | 82       | 0.16      | 159          | 0.20          | 1831       | 1.9         | 920           | 1.5            | 329       | 0.60       |
> | SCE$_{\text{StructComp}}$     | 23       | 0.17      | 59           | 0.16          | 54         | 0.67        | 29            | 1.4            | 16        | 0.62       |
> | COLES                         | 115      | 0.20      | 204          | 0.32          | 1851       | 1.7         | 1018          | 3.1            | 378       | 1.5        |
> | COLES$_{\text{StructComp}}$   | 24       | 0.18      | 60           | 0.21          | 61         | 0.75        | 39            | 1.5            | 21        | 0.64       |
> | GRACE                         | 441      | 1.7       | 714          | 2.5           | 11677      | 18.9        | 5943          | 29.6           | 1996      | 21.2       |
> | GRACE$_{\text{StructComp}}$   | 37       | 0.24      | 72           | 0.87          | 194        | 1.3         | 54            | 2.6            | 59        | 2.2        |
> | CCA-SSG                       | 132      | 1.0       | 225          | 1.1           | 825        | 12.3        | 2418          | 10.5           | 1197      | 11.2       |
> | CCA-SSG$_{\text{StructComp}}$ | 38       | 0.17      | 71           | 0.31          | 85         | 0.9         | 40            | 1.6            | 41        | 1.1        |

---

> ### Author Response · Authors · 2023-11-21
> **Further Response to Reviewer kF3D[2/2]**
>
> As the scale of the graph dataset increases, all GCL models need to use graph sampling or graph partitioning techniques to construct subgraphs and then train GCL on subgraphs. Therefore, pre-processing like METIS is necessary, there are no issue where pre-processing increases the overall time. In our StructComp, METIS only needs to be performed once. Then, we can use the results of the graph partitioning to train various GCL models. *Tables 3 and 4 in the original submission have already proven that StructComp has achieved the best trade-off between performance and efficiency on large graphs.* To further illustrate this point, we provide the overall time results on 100M.
>
> Table 10. The overall time and memory usage on the Ogbn-papers100M dataset.
>
> | Method                        | Acc          | Time (Prepreocess + Training)    | Mem   |
> | ----------------------------- | ------------ | -------------------------------- | ----- |
> | GGD+Cluster-GCN               | 63.5$\pm$0.5 | 1.4h + 1.6h $\times$ 10 =  17.4h | 4.3GB |
> | SCE$_{\text{StructComp}}$     | 63.6$\pm$0.4 | 1.4h + 0.18s$\times$ 10  =  1.4h | 0.1GB |
> | COLES$_{\text{StructComp}}$   | 63.6$\pm$0.4 | 1.4h + 0.16s$\times$ 10 = 1.4h   | 0.3GB |
> | GRACE$_{\text{StructComp}}$   | 64.0$\pm$0.3 | 1.4h + 0.44s$\times$ 10  =  1.4h | 0.9GB |
> | CCA-SSG$_{\text{StructComp}}$ | 63.5$\pm$0.2 | 1.4h + 0.18s$\times$ 10  =  1.4h | 0.1GB |
>
> **Regarding the overhead for inference.** Firstly, the overhead for inference is the same whether or not StructComp is used. The overhead for inference is solely related to the specific encoder used. Secondly, our work is only focused on the training process, as it is the most resource-intensive part of the entire pipeline. In contrast, inference is much simpler compared to training, and there are already straightforward and effective methods available for accelerating inference with GNN encoders [1,2]. For the 100M dataset, we utilized these methods to speed up the inference process. In our experiments, we utilized the same encoder for each dataset.  The corresponding inference overheads are displayed in Table 11.
>
> Table 11. The overhead for inference. Due to the large size of 100M, we employed neighbor sampling during the inference process. For all other datasets, we performed full graph inference.
>
> |      | Cora   | Citeseer | Pubmed | Computers | Photo  | Flickr | Reddit | Arxiv | Products | 100M  |
> | ---- | ------ | -------- | ------ | --------- | ------ | ------ | ------ | ----- | -------- | ----- |
> | Mem  | 24MB   | 63MB     | 122MB  | 72MB      | 40MB   | 362MB  | 1.1GB  | 434MB | 5.9GB    | 5.5GB |
> | Time | 0.001s | 0.001s   | 0.003s | 0.002s    | 0.002s | 0.01s  | 0.03s  | 0.01s | 0.05s    | 159s  |
>
> [1] Hu, W., Fey, M., Zitnik, M., Dong, Y., Ren, H., Liu, B., Catasta, M. and Leskovec, J. Open graph benchmark: Datasets for machine learning on graphs. Neurips 2020.
>
> [2] Gasteiger, J., Qian, C. and Günnemann, S. Influence-based mini-batching for graph neural networks. LOG 2022.

---

> > ### Comment · Reviewer_kF3D · 2023-11-21
> > **Thank you for the reply**
> >
> > Thank you for the follow-up experiments and discussion. I have increased my score accordingly.

---

> > > ### Author Response · Authors · 2023-11-21
> > > **Thank you**
> > >
> > > We are happy to see that we could address your concern. Thank you again for the response and positive comments.

---

### Official Review · Reviewer_iDuQ · 2023-10-31

**Soundness:** 2 fair
**Presentation:** 2 fair
**Contribution:** 2 fair
**Rating:** 5
**Confidence:** 4

**Summary:**

This paper introduces StructComp, a scalable training framework for Graph Contrastive Learning (GCL). By replacing the message-passing operation in GCL with node-compression, StructComp achieves significant reductions in both time and memory consumption. The authors provide both theoretical analysis and empirical evaluations to underscore the effectiveness and efficiency of StructComp in training GCL models.

**Strengths:**

1. The storyline is relatively clear, it is easy to follow for the authors.
2. The experiment results are amazing, especially the time-saving.
3. The used method is quite simple.

**Weaknesses:**

1. Lack of discussion of graph partition: The paper lacks a comprehensive discussion on graph partitioning. Given that the efficacy of the method hinges on graph partitioning—a classic NP-hard problem—a detailed exploration of its impact on the proposed method is warranted. A cursory introduction does not suffice.
2. Inadequate theoretical provements: The theoretical justifications provided are somewhat limited. The authors' attempt to establish the equivalence between the compressed loss and the original loss is based solely on the ER model, which may not be representative of real-world datasets.
3. Lack of the discussion about the limitations.

**Questions:**

1. Adding more discussions about graph partition: The authors should delve deeper into the topic of graph partitioning, as highlighted in the first weakness.

2. Considering not over-claiming your work: It's crucial to avoid overstating the contributions. While the authors assert that they have provided theoretical proof, the strong assumptions (like the ER model) limit its applicability. It might be prudent to either temper such claims in the abstract and introduction or offer more exhaustive proof.
In essence, while I acknowledge the novelty and results presented in this paper, I urge the authors to provide a more in-depth rationale behind their impressive outcomes. Without this, the paper leans more toward a technical report than a comprehensive research paper.

---

> ### Author Response · Authors · 2023-11-16
> **Response to Reviewer iDuQ[1/3]**
>
> Thank you for your comments! Below are our responses.
>
> **Q1:Lack of discussion of graph partition.**
>
> Our main contribution is a novel scalable framework for training GCLs and empirically show that even with off-the-shelf partition algorithms, our framework achieves remarkable speedup. We believe that no need for a specially designed partition algorithm is actually a big advantage.
>
> We agree that different graph partition algorithms will have an impact on the performance of StructComp. We have conducted extra experiments on graph partition. In Table 1, we demonstrate the effects of three algorithms, algebraic JC, variation neighborhoods, and affinity GS, on the performance of StructComp. These three graph coarsening algorithms are widely used in scalable GNNs, from which we can obtain the specific graph partition matrix P. The experimental results suggest that different graph partition methods has little impact on StructComp on these datasets.
>
> Table 1: The results of different graph partition methods.
>
> | Method                            | Cora         | CiteSeer     | PubMed       |
> | --------------------------------- | ------------ | ------------ | ------------ |
> | VN+SCE$_{\text{StructComp}}$      | 81.3$\pm$0.8 | 71.5$\pm$1.0 | 77.5$\pm$2.7 |
> | JC+SCE$_{\text{StructComp}}$      | 81.2$\pm$0.9 | 71.5$\pm$1.1 | 77.3$\pm$2.7 |
> | GS+SCE$_{\text{StructComp}}$      | 81.5$\pm$0.8 | 71.4$\pm$1.0 | 77.4$\pm$3.0 |
> | METIS+SCE$_{\text{StructComp}}$   | 81.6$\pm$0.9 | 71.5$\pm$1.0 | 77.2$\pm$2.9 |
> | VN+COLES$_{\text{StructComp}}$    | 81.4$\pm$0.9 | 71.6$\pm$0.9 | 75.5$\pm$3.0 |
> | JC+COLES$_{\text{StructComp}}$    | 81.4$\pm$0.9 | 71.5$\pm$1.0 | 75.3$\pm$3.0 |
> | GS+COLES$_{\text{StructComp}}$    | 81.8$\pm$0.8 | 71.6$\pm$1.0 | 75.5$\pm$3.2 |
> | METIS+COLES$_{\text{StructComp}}$ | 81.8$\pm$0.8 | 71.6$\pm$0.9 | 75.3$\pm$3.1 |
>
> Graph partition is a common technique for scalable supervised GNN (e.g., ClusterGCN[1], GAS[2]), thus we have not introduced its details in our paper. To our knowledge, Metis is the mainstream graph partitioning method for scalable supervised GNNs. Previous work has not deeply investigated the impact of graph partitioning methods on scalable GNNs. We believe that this is a valuable research topic and we notice that there is an active submission at ICLR 2024 regarding this issue on scalable supervised GNNs .

---

> ### Author Response · Authors · 2023-11-16
> **Response to Reviewer iDuQ[2/3]**
>
> **Q2:Inadequate theoretical provements: The theoretical justifications provided are somewhat limited.**
>
> We understand your concern that the ER model may seem a strong assumption and possibly not entirely reflective of real-world scenarios. However, random graphs like the ER graph and the CSBM is widely adopted for GNN analysis(e.g., [3-8]). In our paper, we choose ER over CSBM since they have no significant difference in unsupervised GCL. We will specify the data distribution we used for analysis in the abstract and introduction. We would also like to point out that strong assumptions are common in the analysis of neural networks, e.g., infinite width in NTK [9], shallow layers [10], removing non-linearity [11], and strong data distribution assumptions [3-8].
>
> According to the suggestion of the reviewer, we give an extra analysis on arbitrary graphs. For non-random graphs, the approximation gap of losses is simply bounded by the Eq 4. Suppose the loss $\mathcal{L}$ is $L$-Lipschitz continuous,
> \begin{equation}
>     \begin{split}
>         |\mathcal{L}(P^\dagger P^TXW)-\mathcal{L}(\hat{A}^kXW)|\leq L \underbrace{\Vert P^\dagger P^T - \hat{A}^k\Vert}_\text{Eq 4.}  \Vert X\Vert  \Vert  W\Vert .
>     \end{split}
> \end{equation}
>
> And for a spectral contrastive loss $\mathcal{L}'$ , assume the graph partition are even, we have:
> \begin{equation}
> \mathcal{L}'(P^TXW)=-\frac{2}{n}\sum_{i=1}^n e^T_{1,i}e_{2,i}+\frac{1}{n^2}\sum_{i=1}^n \sum_{j=1}^n (e^T_{1,i} e_{2,j})^2
>         =-\frac{2}{n}\sum_{k=1}^{n'}\sum_{i\in S_k}e^T_{1,i}e_{2,i}+\frac{1}{n^2}\sum_{i=1}^n \sum_{l=1}^{n'} \sum_{j\in S_l} (e^T_{1,i} e_{2,j})^2
> \end{equation}
> \begin{equation}
>         =-\frac{2}{n'}\sum_{k=1}^{n'}E^T_{1,i}E_{2,i}+\frac{1}{nn'}\sum_{i=1}^n \sum_{l=1}^{n'}  (e^T_{1,i} E_{2,j})^2
>         =-\frac{2}{n'}\sum_{k=1}^{n'}E^T_{1,i}E_{2,i}+\frac{1}{nn'}\sum_{k=1}^{n'}\sum_{i\in S_k} \sum_{l=1}^{n'}  (e^T_{1,i} E_{2,j})^2
> \end{equation}
> \begin{equation}
>         =-\frac{2}{n'}\sum_{k=1}^{n'}E^T_{1,i}E_{2,i}+\frac{1}{n'^2}\sum_{k=1}^{n'}\sum_{l=1}^{n'}  (E^T_{1,i} E_{2,j})^2=\mathcal{L}'(P^\dagger P^TXW),
> \end{equation}
>
> where $e_{1,i}$ denotes the representations of a recovered node and $E^T_{1,i}$ denotes the representations of a compressed node. The above analysis shows that our approximation is reasonable for fixed graphs.
>
> Empirically, we want to highlight *Figure 3 in the original submission* as it greatly validates our approximation on real-world datasets. We have trained two encoders $U$ and $W$ with the compressed loss $\mathcal{L}(X_c; U)$ and the original loss $\mathcal{L}(A,X;W)$, respectively. And we plot the trends of the origianl loss on $U$ and $W$ (i.e., $\mathcal{L}(A,X;U)$ and $\mathcal{L}(A,X;W)$). The figure clearly shows that vanilla training method and StructComp yields extremely similar trends, which justifies our approximation.
>
> **Q3:Lack of the discussion about the limitations.**
>
> Thank you for your feedback. We will provide a detailed discussion of the limitations of StructComp in the revised version. To our knowledge, StructComp is the first training framework specifically designed for GCL models, focusing only on basic and common GCL models. Our future work will involve generalizing StructComp to more complex GCL models, graph data structures.

---

> ### Author Response · Authors · 2023-11-16
> **Response to Reviewer iDuQ[3/3]**
>
> The discussion of all the aforementioned issues will be added into the revised version of our paper. We appreciate your insightful feedback once again.
>
> [1]Chiang, W. L., Liu, X., Si, S., Li, Y., Bengio, S., \& Hsieh, C. J. Cluster-gcn: An efficient algorithm for training deep and large graph convolutional networks. KDD 2019.
>
> [2]Fey, M., Lenssen, J. E., Weichert, F., \& Leskovec, J. Gnnautoscale: Scalable and expressive graph neural networks via historical embeddings. ICML 2021
>
> [3]Wei, R., Yin, H., Jia, J., Benson, A. R., \& Li, P. Understanding non-linearity in graph neural networks from the bayesian-inference perspective. Neurips 2022.
>
> [4]Wu, X., Chen, Z., Wang, W. W., \& Jadbabaie, A. A Non-Asymptotic Analysis of Oversmoothing in Graph Neural Networks. ICLR 2023.
>
> [5]Su, J., Zou, D., Zhang, Z., \& Wu, C. Towards Robust Graph Incremental Learning on Evolving Graphs. ICML 2023.
>
> [6]Keriven, N., Bietti, A., \& Vaiter, S. Convergence and stability of graph convolutional networks on large random graphs. Neurips 2020.
>
> [7]Keriven, N., Bietti, A., \& Vaiter, S. On the universality of graph neural networks on large random graphs. Neurips 2021.
>
> [8]Keriven, N. Not too little, not too much: a theoretical analysis of graph (over) smoothing. Neurips 2022.
>
> [9]Jacot, A., Gabriel, F., \& Hongler, C. Neural tangent kernel: Convergence and generalization in neural networks. Neurips 2018.
>
> [10]Hsu, D., Sanford, C. H., Servedio, R., \& Vlatakis-Gkaragkounis, E. V. On the approximation power of two-layer networks of random relus. COLT 2021.
>
> [11]Awasthi, P., Das, A., \& Gollapudi, S. A convergence analysis of gradient descent on graph neural networks. Neurips 2021.

---

> ### Author Response · Authors · 2023-11-20
> **A friendly reminder for discussion and we appreciate your input**
>
> Dear Reviewer iDuQ,
>
> We thank you again for your insightful and constructive review. We have worked hard and have thoroughly addressed your comments in the rebuttal.
>
> As the discussion period soon comes to an end, we are looking forward to your feedback to our response and revised manuscript. Many thanks again for your time and efforts.
>
> Best regards,
>
> Authors of Submission 5510

---

> ### Author Response · Authors · 2023-11-23
> **A friendly reminder for discussion**
>
> Dear Reviewer iDuQ,
>
> The rebuttal phase ends today and we have not yet received feedback from you. We believe that we have addressed all of your previous concerns. We would really appreciate that if you could check our response and updated paper.
>
> Looking forward to hearing back from you.
>
> Best Regards,
>
> Authors

---

### Official Review · Reviewer_v4Cz · 2023-11-01

**Soundness:** 2 fair
**Presentation:** 3 good
**Contribution:** 2 fair
**Rating:** 5
**Confidence:** 3

**Summary:**

This paper aim at resolving the scalability issue of graph contrastive learning training. In graph representation learning, the most compiutation  overhead comes from message passing, where its complexity grows exponentially wrt the num of layers in GNN.

To overcome this scalability issue, the authors propose **StructComp** trains the encoder with the compressed nodes. StructComp allows the encoder not to perform any message passing during the training stage.

**Strengths:**

I like the idea of using compressing nodes to replace the need of message passing.

**Weaknesses:**

- The theoritical results only hold in linear-GNN, which over-simplifies the problem. It is well know that deep neural network behave different from linear model in contrastive learning [1]. Without consider non-linearity, the problem in Eq. 4 is simply matrix decomposition problem (e.g., [2] section 3).

[1] Understanding the Role of Nonlinearity in Training Dynamics of Contrastive Learning. https://arxiv.org/abs/2206.01342
[2] Understanding Deep Contrastive Learning via Coordinate-wise Optimization https://arxiv.org/pdf/2201.12680.pdf

- Experiment datasets are too small (even arxiv dataset is small)... please try some large-scale graph datasets (e.g., Yelp, Reddit datasets that previously GraphSaint paper) to validate the effectiveness. Especially when this paper is focussing on improving the scalability issue.

- Repeat experiment multiple times instead of just once. For example Figure 4.

**Questions:**

How theoritical results could be generalized to non-linear models?

Does the proposed method work for graphs with multiple node/edge types?

---

> ### Author Response · Authors · 2023-11-16
> **Response to Reviewer v4Cz[1/2]**
>
> Thank you for your detailed review. We would like to address your questions/concerns below:
>
> **Q1: How theoretical results could be generalized to non-linear models?**
>
> Thanks for the insightful question. We explain how to extend the results to non-linear deep models below.
>
> Eq. (4) provides the motivation of structural compression on a linear GNN (which can also be considered as an approximation to one layer in a multi-layer non-linear GNN). The analysis can be extended to non-linear deep GNNs. For instance, given a two-layer non-linear GCN $\sigma(\\hat{A}\sigma(\\hat{A}XW_1)W_2)$, we first approximate $\hat{A}$ by $P^&dagger; P^T$, then the whole GCN can be approximated as
>
> $$
> \begin{align*}
>     \sigma(P^&dagger; P^T\sigma(P^&dagger; P^TXW_1)W_2)&=\sigma(P^&dagger; P^TP^&dagger;\sigma(P^TXW_1)W_2)\\
>     &=P^&dagger;\sigma(\sigma(P^TXW_1)W_2).
> \end{align*}
> $$
>
> The first equality holds because $P^&dagger;$ is a partition matrix and the last equality follows from the fact that $P^TP^&dagger; = I$. Therefore, our analysis provides theoretical justifications of using StructComp as a substitute for non-linear deep GNNs. We will add this extended analysis to the revision. Note that previous studies on scalable methods for training GNNs, such as GraphSaint [1], FastGCN [2], Adapt [3], GRADE [4] also rely on various types of approximation to the propagation matrix, however, their analyses only focus on the approximation quality to a single layer or only work for linear GNNs. On the contrary, our analysis of StructComp can be easily extended to non-linear deep GNNs, which is another advantage of our framework.
>
>
> **Q2:Experiment datasets are too small (even arxiv dataset is small)... please try some large-scale graph datasets (e.g., Yelp, Reddit datasets that previously GraphSaint paper) to validate the effectiveness.**
>
> The ogbn-products dataset used in our experiments has **2,449,029** nodes, which is much larger than both Yelp and Reddit you suggested. We have also conducted extra experiments on Flickr and Reddit to show the performance of StructComp under inductive setting. The experimental results are as follows. StructComp shows remarkable scalability on these datasets.
>
> Table 1: The results on two inductive datasets. OOM means Out of Memory on GPU.
>
> | Method                        | Flickr Acc | Flickr Time | Flickr Mem | Reddit Acc | Reddit Time | Reddit Mem |
> | ----------------------------- | ---------- | ----------- | ---------- | ---------- | ----------- | ---------- |
> | SCE                           | 50.6       | 0.55        | 8427       | -          | -           | OOM        |
> | SCE$_{\text{StructComp}}$     | 51.6       | 0.003       | 43         | 94.4       | 0.017       | 1068       |
> | COLES                         | 50.3       | 0.83        | 9270       | -          | -           | OOM        |
> | COLES$_{\text{StructComp}}$   | 50.7       | 0.003       | 48         | 94.2       | 0.024       | 1175       |
> | GRACE                         | -          | -           | OOM        | -          | -           | OOM        |
> | GRACE$_{\text{StructComp}}$   | 51.5       | 0.010       | 221        | 94.3       | 0.079       | 8683       |
> | CCA-SSG                       | 51.6       | 0.125       | 1672       | 94.9       | 0.21        | 5157       |
> | CCA-SSG$_{\text{StructComp}}$ | 51.8       | 0.007       | 99         | 95.2       | 0.56        | 457        |
>
> According to the suggestions of other reviewers, the experiments on the ogbn-papers100M (which has **111,059,956** nodes) are conducted as well. The experimental results are shown in Table 2. Here, we compressed ogbn-papers100M into a feature matrix $X_c \in R^{5000*128}$ and trained GCL using StructComp. Table 2 also presents the results of GGD trained with ClusterGCN. Although GGD is specifically designed for training large graphs, when dealing with datasets of the scale of ogbn-papers100M, it still requires graph sampling to construct subgraphs and train GGD on a large number of subgraphs. In contrast, our StructComp only requires training a simple and small-scale MLP, resulting in significantly lower resource consumption compared to GGD+ClusterGCN.
>
> Table 2: The accuracy, training time per epoch and memory usage on the Ogbn-papers100M dataset.
>
> | Method                        | Acc          | Time  | Mem   |
> | ----------------------------- | ------------ | ----- | ----- |
> | GGD+ClusterGCN                | 63.5$\pm$0.5 | 1.6h  | 4.3GB |
> | SCE$_{\text{StructComp}}$     | 63.6$\pm$0.4 | 0.18s | 0.1GB |
> | COLES$_{\text{StructComp}}$   | 63.6$\pm$0.4 | 0.16s | 0.3GB |
> | GRACE$_{\text{StructComp}}$   | 64.0$\pm$0.3 | 0.44s | 0.9GB |
> | CCA-SSG$_{\text{StructComp}}$ | 63.5$\pm$0.2 | 0.18s | 0.1GB |

---

> ### Author Response · Authors · 2023-11-16
> **Response to Reviewer v4Cz[2/2]**
>
> **Q3:Repeat experiment multiple times instead of just once. For example Figure 4.**
>
> We emphasize that all of the experiments in the original submission are results of multiple repetitions. On Cora, Citeseer, PubMed, Computers, and Photo, we repeated the experiment 50 times. On Arxiv and Products, we repeated the experiment 5 times. Figure 4 shows the average accuracy of 50 repetitions.
>
> **Q4:Does the proposed method work for graphs with multiple node/edge types?**
>
> Currently, we have not implemented StructComp on graphs with multiple node/edge types, as the chosen basic GCL methods - SCE, COLES, GRACE, and CCA-SSG - are not suitable for these types of graph data. In our future work, we will explore applying StructComp to more complex graph data structures.
>
> --------------------------------
>
> The discussion of all the aforementioned issues will be added into the revised version of our paper. We appreciate your insightful feedback once again.
>
>
>
> [1] H. Zeng, H. Zhou, A. Srivastava, R. Kannan, and V. K. Prasanna. Graphsaint: Graph sampling based inductive learning method. ICLR 2020.
>
> [2] J. Chen, T. Ma, an C. Xiao. Fastgcn: fast learning with graph convolutional networks via importance sampling. ICLR 2018.
>
> [3] W. Huang, T. Zhang, Y. Rong, and J. uang. Adaptive sampling towards fast graph representation learning. NIPS 2018.
>
> [4] R Wang, X Wang, C Shi, L Song. Uncovering the Structural Fairness in Graph Contrastive Learning. Neurips 2022.

---

> ### Author Response · Authors · 2023-11-20
> **A friendly reminder for discussion and we appreciate your input**
>
> Dear Reviewer v4Cz,
>
> We thank you again for your insightful and constructive review. We have worked hard and have thoroughly addressed your comments in the rebuttal.
>
> As the discussion period soon comes to an end, we are looking forward to your feedback to our response and revised manuscript. Many thanks again for your time and efforts.
>
> Best regards,
>
> Authors of Submission 5510

---

> ### Author Response · Authors · 2023-11-21
> **Extra experiments on more complicated graphs**
>
> According to the suggestion from reviewer kF3D, we have conducted experiments to train SP-GCL with StructComp, in order to verify the performance of StructComp on heterophilous graphs. The experimental results are shown in Table 3. Overall, the SP-GCL trained by StructComp is superior to full graph training. This is our initial attempt to use StructComp to handle heterophilous graphs, and it is obviously a valuable direction worth further research.
>
> Table 3. The results on heterophilous datasets.
>
> |                              | Chameleon Acc  | Chameleon Time | Chameleon Mem | Squirrel Acc   | Squirrel Time | Squirrel Mem | Actor Acc      | Actor Time | Actor Mem |
> | ---------------------------- | -------------- | -------------- | ------------- | -------------- | ------------- | ------------ | -------------- | ---------- | --------- |
> | SP-GCL                       | 65.28$\pm$0.53 | 0.038          | 739           | 52.10$\pm$0.67 | 0.080         | 3623         | 28.94$\pm$0.69 | 0.041      | 802       |
> | SP-GCL$_{\text{StructComp}}$ | 66.65$\pm$1.63 | 0.011          | 168           | 53.08$\pm$1.39 | 0.009         | 217          | 28.70$\pm$1.25 | 0.013      | 159       |

---

> ### Author Response · Authors · 2023-11-23
> **A friendly reminder for discussion**
>
> Dear Reviewer v4Cz,
>
> The rebuttal phase ends today and we have not yet received feedback from you. We believe that we have addressed all of your previous concerns. We would really appreciate that if you could check our response and updated paper.
>
> Looking forward to hearing back from you.
>
> Best Regards,
>
> Authors

---

### Official Review · Reviewer_PXMh · 2023-11-01

**Soundness:** 2 fair
**Presentation:** 2 fair
**Contribution:** 2 fair
**Rating:** 5
**Confidence:** 3

**Summary:**

The paper proposes Structural Compression (StructComp), a new training framework that improves the scalability of graph contrastive learning (GCL) models. The key idea is to substitute propagation with a sparse, low-rank approximation of the diffusion matrix to compress the nodes. Contrastive learning is performed on these compressed nodes, reducing computation and memory costs. Theoretical analysis shows the compressed loss approximates the original loss and StructComp implicitly regularizes the model.  Experiments on various single-view and multi-view GCL methods demonstrate StructComp's improvements in performance and efficiency.

**Strengths:**

1.	The paper is well-written and easy to follow. The problem is motivated well, and the method is explained clearly.

2.	Scalability is a major bottleneck hindering wider adoption of graph neural networks. This work makes an important contribution by enabling efficient training of GCL models.

**Weaknesses:**

1. Additional experiments could help verify claims on scalability and robustness of StructComp:

- Evaluating on larger datasets like papers100M and or OGBG-LSC datasets  would better support scalability claims, since the experimented datasets are rather small or medium scale.

- Would be great to verify the model stability/robustness with the proposed regularization, since it is claimed in the presentation.


2. Would be great to discuss the approximation quality to the diffusion matrix  of StructComp for more complicated graphs and other models architectures (like GAT, GraphSAGE) .

3. There is a lack of  comparisons with certain related works, such as recent graph contrastive learning methods  [1-2]


[1] ] Wang, H., Zhang, J., Zhu, Q., & Huang, W. (2022). Can Single-Pass Contrastive Learning Work for Both Homophilic and Heterophilic Graph?. arXiv preprint arXiv:2211.10890.

[2] Li, J., Sun, W., Wu, R., Zhu, Y., Chen, L., & Zheng, Z. (2023). Scaling Up, Scaling Deep: Blockwise Graph Contrastive Learning. arXiv preprint arXiv:2306.02117.

**Questions:**

See the weakness above

---

> ### Author Response · Authors · 2023-11-16
> **Response to Reviewer PXMh[1/3]**
>
> Thank you for your detailed review. We would like to address your questions/concerns below:
>
> **Q1:Evaluating on larger datasets like papers100M and or OGBG-LSC datasets would better support scalability claims, since the experimented datasets are rather small or medium scale.**
>
> We have conducted extra experiments on the ogbn-papers100M dataset according to your suggestion. We use StructComp to train four representative GCL models. Here, we compressed ogbn-papers100M into a feature matrix $X_c \in R^{5000*128}$ and trained GCL using StructComp. The table also presents the results of GGD trained with ClusterGCN. Although GGD is specifically designed for training large graphs, when dealing with datasets of the scale of ogbn-papers100M, it still requires graph sampling to construct subgraphs and train GGD on a large number of subgraphs. In contrast, our StructComp only requires training a simple and small-scale MLP, resulting in significantly lower resource consumption compared to GGD+ClusterGCN.
>
>
>
> Table 1: The accuracy, training time per epoch and memory usage on the Ogbn-papers100M dataset.
>
> | Method                        | Acc          | Time  | Mem   |
> | ----------------------------- | ------------ | ----- | ----- |
> | GGD                           | 63.5$\pm$0.5 | 1.6h  | 4.3GB |
> | SCE$_{\text{StructComp}}$     | 63.6$\pm$0.4 | 0.18s | 0.1GB |
> | COLES$_{\text{StructComp}}$   | 63.6$\pm$0.4 | 0.16s | 0.3GB |
> | GRACE$_{\text{StructComp}}$   | 64.0$\pm$0.3 | 0.44s | 0.9GB |
> | CCA-SSG$_{\text{StructComp}}$ | 63.5$\pm$0.2 | 0.18s | 0.1GB |
>
>
>
> **Q2:Would be great to verify the model stability/robustness with the proposed regularization, since it is claimed in the presentation.**
>
> We have conducted extra experiments to study the robustness of StructComp. We randomly add 10\% of noisy edges into three datasets and perform the node classification task. On the original dataset, the models trained with StructComp showed performance improvements of **0.36**, **0.40**, **1.30** and **1.87**, respectively, compared to the models trained with full graphs. With noisy perturbation, the models trained with StructComp showed performance improvements of **0.80**, **1.27**, **2.47**, and **1.87**, respectively, compared to full graph training. This indicates that GCL models trained with StructComp exhibit better robustness.
>
>
>
> Table 2: The results over 50 random splits  on the perturbed datasets.
>
> | Method                        | Cora         | CiteSeer     | PubMed       |
> | ----------------------------- | ------------ | ------------ | ------------ |
> | SCE                           | 78.8$\pm$1.2 | 69.7$\pm$1.0 | 73.4$\pm$2.2 |
> | SCE$_{\text{StructComp}}$     | 79.3$\pm$0.9 | 69.3$\pm$0.9 | 75.7$\pm$2.8 |
> | COLES                         | 78.7$\pm$1.2 | 68.0$\pm$1.0 | 66.5$\pm$1.8 |
> | COLES$_{\text{StructComp}}$   | 79.0$\pm$1.0 | 68.3$\pm$0.9 | 69.7$\pm$2.6 |
> | GRACE                         | 77.6$\pm$1.1 | 64.1$\pm$1.4 | 64.5$\pm$1.7 |
> | GRACE$_{\text{StructComp}}$   | 78.3$\pm$0.8 | 69.1$\pm$0.9 | 66.2$\pm$2.4 |
> | CCA-SSG                       | 75.5$\pm$1.3 | 69.1$\pm$1.2 | 73.5$\pm$2.2 |
> | CCA-SSG$_{\text{StructComp}}$ | 78.2$\pm$0.7 | 69.2$\pm$0.8 | 76.3$\pm$2.5 |

---

> ### Author Response · Authors · 2023-11-16
> **Response to Reviewer PXMh[2/3]**
>
> **Q3:Would be great to discuss the approximation quality to the diffusion matrix of StructComp for more complicated graphs and other models architectures (like GAT, GraphSAGE).**
>
> In order to verify the approximation quality to the diffusion matrix of StructComp, we test the performance on a deep GNN architecture called SSGC [1]. We transferred the trained parameters of StructComp to the SSGC encoder for inference. For full graph training in GCL, both the training and inference stages were performed using the SSGC encoder. Table 3 shows our experimental results, indicating that even with a deeper and more complicated encoder, StructComp still achieved outstanding performance. To the best of our knowledge, current mainstream GCL models, whether single-view models such as SCE, COLES, GLEN [2], or multi-view models such as DGI, GRACE, CCA-SSG, GGD, and their variants, all use SGC or GCN encoders. These unsupervised GCL models can achieve performance surpassing supervised GNNs (such as GAT, SAGE) with simple encoders on many datasets. Few GCL models have focused on whether using different encoders can achieve better performance. Moreover, GCL training is more complicated than supervised GNNs, so introducing complex encoders may make the models more difficult to train. Considering this, we did not extensively explore how using other encoders would affect StructComp. We believe this is a good question, and we will investigate it further in future work.
>
>
>
> Table 3: The results of GCLs with SSGC encoders over 50 random splits.
>
> | Method                        | Cora         | Citeseer     | Pubmed       |
> | ----------------------------- | ------------ | ------------ | ------------ |
> | SCE                           | 81.8$\pm$0.9 | 72.0$\pm$0.9 | 78.4$\pm$2.8 |
> | SCE$_{\text{StructComp}}$     | 82.0$\pm$0.8 | 71.7$\pm$0.9 | 77.8$\pm$2.9 |
> | COLES                         | 81.8$\pm$0.9 | 71.3$\pm$1.1 | 74.8$\pm$3.4 |
> | COLES$_{\text{StructComp}}$   | 82.0$\pm$0.8 | 71.6$\pm$1.0 | 75.6$\pm$3.0 |
> | GRACE                         | 80.2$\pm$0.8 | 70.7$\pm$1.0 | 77.3$\pm$2.7 |
> | GRACE$_{\text{StructComp}}$   | 81.1$\pm$0.8 | 71.0$\pm$1.0 | 78.2$\pm$1.3 |
> | CCA-SSG                       | 82.1$\pm$0.9 | 71.9$\pm$0.9 | 78.2$\pm$2.8 |
> | CCA-SSG$_{\text{StructComp}}$ | 82.6$\pm$0.7 | 71.7$\pm$0.9 | 79.4$\pm$2.6 |

---

> ### Author Response · Authors · 2023-11-16
> **Response to Reviewer PXMh[3/3]**
>
> **Q4:There is a lack of comparisons with certain related works, such as recent graph contrastive learning methods [1-2].**
>
> It should be noted that the goal of these studies and our work are different. The aim of SPGCL is to handle homophilic graphs and heterophilic graphs simultaneously. BlockGCL attempts to explore the application of deep GNN encoder in the GCL field. On the other hand, StructComp is a framework designed to scale up the training of GCL models: it aims to efficiently train common GCL models without performance drop. It is not a new GCL model that aims to achieve  SOTA performance compared to existing GCL models. So our work is orthogonal to these two previous works. In fact, StructComp can be used as the training method for both SP-GCL and BlockGCL. In future work, we will further investigate how to train these recent graph contrastive learning methods using StructComp.
>
> In terms of scalability, which is the main goal of our work, we have conducted extra experiments to compare SP-GCL, BlockGCL and StructComp-trained baseline. The results confirm that the aforementioned two GCL models are not designed for reducing training costs.
>
>
>
> Table 4: The results of StructComp-trained GCLs and some GCL baselines over 50 random splits. For SP-GCL, we are unable to get the classification accuracy on CiteSeer since it does not take isolated nodes as input.
>
> | Method                                                       | Cora Acc     | Cora Time | Cora Mem | CiteSeer Acc | CiteSeer Time | CiteSeer Mem | PubMed Acc   | PubMed Time | PubMed Mem |
> | ------------------------------------------------------------ | ------------ | --------- | -------- | ------------ | ------------- | ------------ | ------------ | ----------- | ---------- |
> | BlockGCL                                                     | 78.1$\pm$2.0 | 0.026     | 180      | 64.5$\pm$2.0 | 0.023         | 329          | 74.7$\pm$3.1 | 0.037       | 986        |
> | SP-GCL                                                       | 81.4$\pm$1.2 | 0.016     | 247      | -            | 0.021         | 319          | 74.8$\pm$3.2 | 0.041       | 1420       |
> | SCE$_{\text{StructComp}}$                                    | 81.6$\pm$0.9 | 0.002     | 23       | 71.5$\pm$1.0 | 0.002         | 59           | 77.2$\pm$2.9 | 0.003       | 54         |
> | COLES$_{\text{StructComp}}$                                  | 81.8$\pm$0.8 | 0.002     | 24       | 71.6$\pm$0.9 | 0.003         | 60           | 75.3$\pm$3.1 | 0.003       | 61         |
> | GRACE$_{\text{StructComp}}$                                  | 79.7$\pm$0.9 | 0.009     | 37       | 70.5$\pm$1.0 | 0.009         | 72           | 77.2$\pm$1.4 | 0.009       | 194        |
> | CCA-SSG$_{\text{StructComp}}$ &nbsp;&nbsp;&nbsp;&nbsp;&nbsp;&nbsp; &nbsp; | 82.3$\pm$0.8 | 0.006     | 38       | 71.6$\pm$0.9 | 0.005         | 71           | 78.3$\pm$2.5 | 0.006       | 85         |
>
>
>
> ------------------------------------------------------------
>
> The discussion of all the aforementioned issues will be added into the revised version of our paper. We appreciate your insightful feedback once again.
>
>
>
> [1]H. Zhu, and P. Koniusz. Simple spectral graph convolution. ICLR 2020.
>
> [2]H. Zhu, and P. Koniusz. Generalized Laplacian Eigenmaps. Neurips 2022.

---

> ### Author Response · Authors · 2023-11-20
> **A friendly reminder for discussion and we appreciate your input**
>
> Dear Reviewer PXMh,
>
> We thank you again for your insightful and constructive review. We have worked hard and have thoroughly addressed your comments in the rebuttal.
>
> As the discussion period soon comes to an end, we are looking forward to your feedback to our response and revised manuscript. Many thanks again for your time and efforts.
>
> Best regards,
>
> Authors of Submission 5510

---

> ### Author Response · Authors · 2023-11-21
> **Extra experiments on more complicated graphs**
>
> According to the suggestion from reviewer kF3D, we have conducted experiments to train SP-GCL with StructComp, in order to verify the performance of StructComp on heterophilous graphs. The experimental results are shown in Table 5. Overall, the SP-GCL trained by StructComp is superior to full graph training. This is our initial attempt to use StructComp to handle heterophilous graphs, and it is obviously a valuable direction worth further research.
>
> Table 5. The results on heterophilous datasets.
> |                              | Chameleon Acc  | Chameleon Time | Chameleon Mem | Squirrel Acc   | Squirrel Time | Squirrel Mem | Actor Acc      | Actor Time | Actor Mem |
> | ---------------------------- | -------------- | -------------- | ------------- | -------------- | ------------- | ------------ | -------------- | ---------- | --------- |
> | SP-GCL                       | 65.28$\pm$0.53 | 0.038          | 739           | 52.10$\pm$0.67 | 0.080         | 3623         | 28.94$\pm$0.69 | 0.041      | 802       |
> | SP-GCL$_{\text{StructComp}}$ | 66.65$\pm$1.63 | 0.011          | 168           | 53.08$\pm$1.39 | 0.009         | 217          | 28.70$\pm$1.25 | 0.013      | 159       |

---

> ### Author Response · Authors · 2023-11-23
> **A friendly reminder for discussion**
>
> Dear Reviewer PXMh,
>
> The rebuttal phase ends today and we have not yet received feedback from you. We believe that we have addressed all of your previous concerns. We would really appreciate that if you could check our response and updated paper.
>
> Looking forward to hearing back from you.
>
> Best Regards,
>
> Authors

---

### Meta-Review · Area_Chair_FYXv · 2023-12-10

**Metareview:**

Paper proposes a technique called Structural Compression to improve the compute and memory efficiency for Graph Contrastive Learning (GCL). Main idea is to merge clusters of nodes into a super-nodes and perform GCL on the  “super-graph” of super-nodes (done using standard min-cut algorithms). Super-node embedding is taken as the average node embedding of cluster components. Authors provide experimental results showing efficiency and performance gains. Authors argue that performance gain is due to regularization effect of the compression. Additionally they also provide a new augmentation method for multi-view GCL with Structural Compression. During the review process authors were able to provide mostly positive additional experiments on robustness, larger graphs, heterophillc graphs, and other baselines.

However, there is very no discussion (even after asked during open discussion) on the limitation of the method, and it is not clear why it should work for heterophillic graphs (note it reduced performance in one such dataset). Nitpicking: $^\dagger$ is not a good notation since it is traditionally being used for pseudo inverse of a matrix.

**Justification For Why Not Higher Score:**

Questions on limitation and heterophilic graphs.

**Justification For Why Not Lower Score:**

Novel simple idea and good results

---

### Decision · Program_Chairs · 2024-01-16

Accept (poster)